# Mitochondria regulate inositol triphosphate-mediated Ca²⁺ release triggered by voltage-dependent Ca²⁺ entry in resistance arteries

Xun Zhang ⓘ, Charlotte Buckley ⓘ, Matthew D. Lee ⓘ, Susan Chalmers ⓘ, Calum Wilson ⓘ and John G. McCarron ⓘ

*Strathclyde Institute of Pharmacy and Biomedical Sciences, University of Strathclyde, Glasgow, UK*

Handling Editors: Bjorn Knollmann & Nikki Jernigan

The peer review history is available in the Supporting Information section of this article (https://doi.org/10.1113/JP288022#support-information-section).

**Abstract figure legend** Proposed mechanism of regulation of mitochondria on VDCCs mediated Ca²⁺ signals in smooth muscle cells. Depolarization of the plasma membrane induces Ca²⁺ influx via L-type VDCCs that produces a steady elevation in Ca²⁺ and activates IP₃Rs to evoke repetitive Ca²⁺ release events from the intracellular Ca²⁺ store. Depolarization of mitochondrial membrane potential reduces IP₃R activity. Similarly inhibition of the ATP synthase increases ROS production, which also decreases IP₃R activity. Ca²⁺ entry via VDCCs is unaffected by changes in the mitochondrial membrane potential or inhibition of the ATP synthase. IP₃Rs, inositol phosphate receptors; ROS, reactive oxidative species; RyRs, ryanodine receptors; SR, sarcoplasmic reticulum; VDCCs, voltage-dependent Ca²⁺ channels.

**Xun Zhang** is a research associate in the Strathclyde Institute of Pharmacology and Biomedical Sciences, University of Strathclyde. Xun joined John G. McCarron's lab after obtaining his PhD from Glasgow Caledonian University. Xun's research interest focuses on clarifying the vital roles of Ca²⁺ signalling and mitochondria in the regulation of physiological functions of vascular tissues, including endothelial and smooth muscle cells using cutting-edge microscopy. His long-term goal is to understand the disease mechanisms underlying chronic cardiovascular disorders, for example, hypertension and diabetes, and discover potent therapeutics. Xun is also a member of The Physiological Society and a reviewer for peer-reviewed journals, for example, *Frontier in Immunology*.

The Journal of Physiology

**Abstract** An increase in cytoplasmic $Ca^{2+}$ concentration activates multiple cellular activities, including cell division, metabolism, growth, contraction and death. In smooth muscle $Ca^{2+}$ entry via voltage-dependent $Ca^{2+}$ channels leads to a relatively uniform increase in cytoplasmic $Ca^{2+}$ levels that facilitates co-ordinated contraction throughout the cell. However certain functions triggered by voltage-dependent $Ca^{2+}$ channels require periodic, pulsatile $Ca^{2+}$ changes. The mechanism by which $Ca^{2+}$ entry through voltage-dependent channels supports both co-ordinated contraction and distinct cellular responses driven by pulsatile $Ca^{2+}$ changes is unclear. Here in intact resistance arteries we show that $Ca^{2+}$ entry via voltage-dependent $Ca^{2+}$ channels evokes $Ca^{2+}$ release via inositol triphosphate receptors ($IP_3Rs$), generating repetitive $Ca^{2+}$ oscillations and waves. We also show that mitochondria play a vital role in regulating $Ca^{2+}$ signals evoked by voltage-dependent $Ca^{2+}$ entry by selectively modulating $Ca^{2+}$ release via $IP_3Rs$. Depolarizing the mitochondrial membrane inhibits $Ca^{2+}$ release from internal stores, reducing the overall signal-generated $Ca^{2+}$ influx without altering the signal resulting from voltage-dependent $Ca^{2+}$ entry. Notably neither $Ca^{2+}$ entry via voltage-dependent $Ca^{2+}$ channels nor $Ca^{2+}$ release via $IP_3Rs$ alters mitochondrial location or mitochondrial membrane potential in intact smooth muscle cells. Collectively these results demonstrate that activation of voltage-dependent $Ca^{2+}$ channels drives $Ca^{2+}$ entry, which subsequently triggers $Ca^{2+}$ release from the internal store in smooth muscle cells. Mitochondria selectively regulate this process by modulating $IP_3R$-mediated amplification of $Ca^{2+}$ signals, ensuring that different cellular responses are precisely controlled.

(Received 31 October 2024; accepted after revision 19 March 2025; first published online 17 April 2025)

**Corresponding author** J. G. McCarron: Strathclyde Institute of Pharmacy and Biomedical Sciences, University of Strathclyde, 161 Cathedral Street, Glasgow G4 0RE, UK. Email: john.mccarron@strath.ac.uk

**Key points**

- In smooth muscle $Ca^{2+}$ entry via voltage-dependent channels produces a uniform $Ca^{2+}$ increase, enabling co-ordinated contraction in each cell.
- Certain functions, however, require large, pulsatile $Ca^{2+}$ changes rather than a uniform increase.
- Using advanced imaging in intact arteries, we discovered that voltage-dependent $Ca^{2+}$ entry triggers internal store $Ca^{2+}$ release via $IP_3$ receptors, generating repetitive $Ca^{2+}$ oscillations and waves.
- Mitochondria selectively modulate these signals by regulating only $IP_3$ receptor-mediated release; neither mitochondrial location nor membrane potential is altered by either type of $Ca^{2+}$ signal.
- These findings demonstrate how voltage-dependent $Ca^{2+}$ entry supports both co-ordinated contraction and pulsatile $Ca^{2+}$-driven biological responses.

## Introduction

The contractility of vascular smooth muscle cells in resistance arteries (diameter $\sim <200$ μm) is a key determinant of blood pressure regulation (Brozovich et al., 2016; Touyz et al., 2018). A major trigger for smooth muscle contraction is an increase in cytoplasmic $Ca^{2+}$ concentration, primarily as a result of influx of the ion across the cell membrane. Among the various pathways for $Ca^{2+}$ entry, the dihydropyridine-sensitive voltage-dependent $Ca^{2+}$ channels (VDCCs) are widely expressed and may play a dominant role in controlling $Ca^{2+}$ influx in smooth muscle cells (Berridge, 2008; Sanders, 2001). The critical role of these channels in vascular function is highlighted by the widespread use of $Ca^{2+}$ channel blockers, including dihydropyridines, in blood pressure management (Xu et al., 2016).

In addition to controlling contraction $Ca^{2+}$ influx via VDCCs regulates nearly all cellular activities, including division, growth and death. This broad functional impact results from the ability of cells to generate complex spatiotemporal signals that drive distinct cellular responses (McCarron et al., 2006). However the mechanisms underlying the generation of different types of $Ca^{2+}$ signals is not fully understood.

The complexity of $Ca^{2+}$ control is highlighted by the regulation of smooth muscle contraction. A relatively uniform increase in cytoplasmic $Ca^{2+}$ concentration is

required to produce a co-ordinated contraction, whereas other cell functions rely on pulsatile $Ca^{2+}$ changes. This raises an important question: how can $Ca^{2+}$ entry through VDCCs, which co-ordinate a sustained contraction, also selectively target additional discrete cellular responses that require pulsatile $Ca^{2+}$ changes? A partial answer lies in the ability of $Ca^{2+}$ entering the cell to interact with other intracellular signalling elements to make the cytoplasm 'excitable'. This excitability leads to repetitive pulsatile signalling events (oscillations) and complex spatiotemporal $Ca^{2+}$ signals, including propagating $Ca^{2+}$ waves that transmit information within and between cells (McCarron et al., 2002; Pérez et al., 2001).

The ability of cells to produce oscillations and waves offers several advantages for selective $Ca^{2+}$ signalling. For example certain enzymes, like calmodulin kinase II and protein kinase C, respond to the frequency of $Ca^{2+}$ oscillations rather than to steady-state increases in cytoplasmic $Ca^{2+}$ levels. In Jurkat T-cells $Ca^{2+}$ oscillations selectively activate transcription factors and gene expression, a process likely to occur in smooth muscle cells as well. Despite their importance the precise events that trigger $Ca^{2+}$ oscillations and govern $Ca^{2+}$ wave propagation remain unclear. Various hypotheses have been proposed, but none have been universally accepted. Two prominent proposals focus on the receptor complexes on the sarcoplasmic reticulum (SR) that control $Ca^{2+}$ release, specifically the inositol triphosphate receptor ($IP_3R$) and the ryanodine receptor (RyR).

Intracellular $Ca^{2+}$ signals in smooth muscle, whether originating from the release of $Ca^{2+}$ by the SR or from influx of the ion across the plasma membrane, are influenced by other cellular organelles, particularly mitochondria. This mitochondrial control provides an additional layer of complexity to the spatial and temporal patterns of $Ca^{2+}$ signalling. Although mitochondria are primarily known for producing ATP through oxidative phosphorylation, they also serve as dynamic $Ca^{2+}$ buffers. As a result they play a critical role in regulating $Ca^{2+}$ signalling in native smooth muscle (Berridge, 2008; Drummond et al., 2000; McCarron & Muir, 1999).

Mitochondrial $Ca^{2+}$ uptake is driven by the inside-negative mitochondrial electrical gradient ($\Delta\Psi m$) and occurs when the cytoplasmic $Ca^{2+}$ concentration exceeds 0.2–1 μM (Bailey et al., 2005; Becker et al., 1980; Heaton & Nicholls, 1976; Nicholls & Crompton, 1980; Pitter et al., 2002). This uptake can modulate the rate of change of $Ca^{2+}$ in the cytoplasm and result in feedback control of SR $Ca^{2+}$ release or $Ca^{2+}$ influx. Additionally mitochondria may modulate $Ca^{2+}$ signalling through the provision of ATP or production of reactive oxygen species (ROS) as each exerts influence on SR $Ca^{2+}$ release and $Ca^{2+}$ influx. However the precise contribution of these mitochondrial mechanisms to $Ca^{2+}$ signalling in native smooth muscle is not fully understood.

In addition to the mitochondrial electrical gradient, the morphology of the organelle may be important in the ability of mitochondria to influence the development of spatiotemporal $Ca^{2+}$ signalling patterns. Some cell types, such as fibroblasts or astrocytes, have an extended, continuous mitochondrial network that branches throughout the cell (Diaz et al., 2000; Marchant et al., 2002; Rizzuto et al., 1998). In such cases even localized mitochondrial activity can have cell-wide consequences (Skulachev, 2001), for example, through internal shuttling of $Ca^{2+}$ within the mitochondrial network. In other cells the morphology is complex, and mitochondria exist in a variety of forms because the organelle exhibits dynamic behaviour and may almost constantly change shape and move via Brownian motion, stochastically determined directed motion, long-range motor-driven displacement, fission and fusion (Miller & Sheetz, 2004, 2006; Saunter et al., 2009). However in native smooth muscle, and many other native cell types, mitochondria exist as discrete ovoid structures ∼1–5 μm in length (Chalmers et al., 2015; Collins et al., 2002; Dai et al., 2005; Duchen et al., 1998; Ichas et al., 1997; O'Reilly et al., 2003; Restini et al., 2006). This type of structure may lend itself to mitochondria acting as independent entities.

The structure of the organelle is important in determining the influence of mitochondria on $Ca^{2+}$ signalling, and mitochondria may act as a network even if not electrically connected. For example exposure to apoptotic agents can trigger accumulation of $Ca^{2+}$ by mitochondria, initiating a regenerative wave of mitochondrial depolarization that spreads throughout the entire mitochondrial complement. This, in turn, induces a wave of cytoplasmic $Ca^{2+}$ release from mitochondria (Pacher & Hajnóczky, 2001). Clearly, in response to some $Ca^{2+}$ signals, the organization of mitochondria is critical in the organelle's ability to influence cellular activity.

Much of our understanding of voltage-dependent $Ca^{2+}$ entry and mitochondrial contributions to $Ca^{2+}$ signalling in smooth muscle comes from studies on cultured smooth muscle or isolated single smooth muscle cells. However these models have limitations. Cultured smooth muscle cells change drastically from the native cell type so that their physiological relevance is not always clear. For example mitochondria are highly dynamic in cultured smooth muscle cells but are largely immobile in native smooth muscle (Chalmers et al., 2012). Although physiological function is preserved in freshly isolated native smooth muscle cells, they lack intercellular inter-actions found in intact tissue, which may alter signalling responses.

In the present study the behaviour of individual smooth muscle cells, within intact small arteries, has been examined to determine the mechanisms that under-lie the intracellular $Ca^{2+}$ signalling cascade triggered by voltage-dependent $Ca^{2+}$ entry. Additionally we explored

the role of mitochondria in regulating $Ca^{2+}$ dynamics in these native smooth muscle cells.

## Methods

### Animals

Adult male SD (Sprague–Dawley) IGS rats (10–12 weeks old) (Charles River, UK) were used in this study. Three animals were housed per cage (RC2F cages, North Kent Plastics Company, Leicester, UK) in an environment enriched with aspen wood chew sticks, hanging huts and nesting materials (Sizzle nest, LBS Technology, London, UK). Animals had access to fresh water and chow (RM1, Special Diet Service, Romford, UK) *ad libitum*. Housing environment was set at room temperature (19–23°C, set point 21°C), a humidity of 45%–65% and a 12-h light cycle. All animal care and experimental procedures were carried out in compliance with the University of Strathclyde Animal Welfare and Ethical Review Board (Schedule 1 procedure; Animals (Scientific Procedures) Act 1986, UK), under UK Home Office regulations. Animals were killed by cervical dislocation, with death confirmed by exsanguination. After the animals were killed, the whole mesentery bed was rapidly dissected and transferred to a dissecting dish filled with a physiological salt solution (PSS) with the following composition (in mM): 145.0 NaCl, 2.0 MOPS, 4.7 KCl, 1.2 $NaH_2PO_4$, 5.0 glucose, 0.02 EDTA, 1.17 $MgCl_2$, and 2.0 $CaCl_2$ (pH adjusted to 7.4 with NaOH).

### Chemicals and solutions

Cal520 (ab171868) was obtained from Abcam (Cambridge, UK). Pluronic F-127 and caged-Ins (1,4,5) $P_3$/PM (cag-iso-2-145-100, caged ($cIP_3$)) were obtained from Sichem (Bremen, Germany). Tetramethylrhodamine ethyl ester perchlorate (TMRE, 87917) was obtained from Sigma-Aldrich (Dorset, UK). The ROS-ID Total ROS/Superoxide Detection Kit was obtained from Enzo Life Sciences (ENZ-51010; Exeter, UK). Sylgard 184 silicone elastomer kit was obtained from Dow (Barry, UK).

Several pharmacological tools were used in this study. Oligomycin (O4876), carbonyl cyanide *m*-chlorophenylhydrazone (CCCP, 215911), Mito TEMPO (SML0737), 2-aminoethyl diphenylborinate (2-APB, D9754), caffeine (C-0750), U73122 (U6756) and DMSO (D8418) were obtained from Sigma-Aldrich (Dorset, UK). Dantrolene (0507) and nimodipine (0600) were obtained from Bio-techne (Abingdon, UK). Ryanodine (HB1320) was sourced from HelloBio (Bristol, UK).

PSS consisted of the following chemicals: NaCl (S7653), MOPS (M5162), $NaH_2PO_4$ (S9638), glucose (G7528), EDTA (03690), EGTA (E4378), $CaCl_2$ (21115) and $MgCl_2$ (63069); these chemicals were obtained from Sigma-Aldrich (Dorset, UK). KCl (7447-40-7) was obtained from VWR (Radnor, Pennsylvania, USA).

For immunostaining assays the following reagents were used: phosphate-buffered saline (PBS, P4417), Triton X-100 (93443), glycine (G7126), bovine serum albumin (BSA, A3294), donkey serum (D9663) and 4',6-diamidino-2-phenylindole (DAPI; D9542, diluted to 1 ng/ml); all these were obtained from Sigma-Aldrich (Dorset, UK). Paraformaldehyde (18505) was obtained from Ted Pella (Redding, California, USA). Primary antibodies included anti-Cav1.2 (MA5-27717, 1:200 dilution) and actin-GFP (C10582) from Thermo Fisher Scientific (Cambridge, UK) and anti-$IP_3$Rs (238893, 1:100 dilution) from Millipore (Livingston, UK). Secondary antibodies, including donkey anti-rabbit IgG Alexa Fluor 555 (A-31572), donkey anti-mouse IgG Alexa Fluor 488 (A-21202) and anti-goat IgG Alexa Fluor 488 (A-11055), were purchased from Life Technologies (Paisley, UK).

### Imaging systems

Depending on the experimental requirements we used one of three different imaging systems to visualize smooth muscle cell signalling. Each system was equipped with a custom bath chamber and subjected to superfusion using a Gilson minipuls3 pump (Dunstable, UK) connected via PVC Tygon S3 E-3603 (FisherScientific, Paisley, UK) and peristaltic pump tubing (Gilson, Dunstable, UK).

**Imaging system 1 (widefield fluorescence).** Images of $Ca^{2+}$ signals and immunostaining were obtained using a Nikon FN-1 (Amstelveen, Netherlands) upright fluorescence microscope equipped with a 40× (0.8 numerical aperture) or 16× (0.8 numerical aperture) water dipping lens. A CoolLED pE-4000 illumination system was used (Andover, UK), and the camera used was a back-illuminated electron-multiplying charge-coupled device (EMCCD) (1024 × 1024 13 µm pixels, iXon 888, Andor, Belfast, UK). Images were obtained at 10 Hz using µManager software (Edelstein et al., 2010, 2014).

**Imaging system 2 (photolysis-triggered $Ca^{2+}$ imaging).** Images of $Ca^{2+}$ signals initiated via the photolysis of $cIP_3$ was obtained using a Nikon FN-1 microscope equipped with a Rapp Optoelectronics (Wedel, Germany) photomanipulation system. The system consisted of a DL-Series UV (375 nm) laser coupled to a UGA-42 Firefly scanner, allowing the uncaging region to be set via software. The UV photolysis light was first passed through an attenuating neutral density filter (1% transmission) and

used at a power of 2 mW before optical losses. Images were obtained at 10 Hz using a Photometrics Evolve 13 EMCCD camera and a CoolLED pT-100 illumination system with µManager software.

**Imaging system 3 (dual-channel confocal imaging).** Dual-channel imaging of Ca$^{2+}$ signals (excitation wavelength: 488 nm) and mitochondrial membrane potential (excitation wavelength: 561 nm) was conducted using a Nikon Ti2 microscope equipped with an AX R confocal scan head, a Plan Apo Lambda S 25XC Sil lens (1.05 numerical aperture), and an NSPARC camera (Nikon).

### Immunohistochemistry

Freshly isolated arteries were cut open and pinned onto a Sylgard-coated six-well plate. Endothelial cells were then removed using a strand of human hair. The arteries were then fixed with 4% paraformaldehyde (PFA) for 15 min (twice). After fixation arteries were washed thrice in glycine solution (0.75% in H$_2$O$_2$) for 5 min to neutralize residual PFA, then washed with PBS (thrice) and permeabilized using 0.2% Triton X-100 for 30 min. Triton X-100 was subsequently washed with PBS (thrice) and then with an antibody wash solution (0.2% Triton X-100 in PBS, thrice).

To block non-specific binding sites arteries were incubated in blocking buffer (5% BSA, 5% donkey serum in an antibody wash solution) for 1 h and then rinsed thrice with an antibody wash solution. Arteries were incubated overnight at 4°C on an orbital shaker (Stuart, UK) with primary antibodies (anti-Cav1.2, 1:200 dilution; anti-IP$_3$Rs, 1:100 dilution). Primary antibodies were diluted in an antibody buffer solution (1% BSA, 1% donkey serum in an antibody wash buffer). As a negative control some arteries were incubated with the antibody buffer solution alone. After incubation primary antibodies were washed using the antibody wash solution for 5 min (thrice). Preparations were then incubated with secondary antibodies for 1 h at room temperature (donkey anti-mouse IgG Alexa Fluor 488 for Cav1.2, donkey anti-rabbit IgG Alexa Fluor 555 for IP$_3$Rs with final concentration: 2 µg/ml). After incubation secondary antibodies were washed with the antibody wash solution for 5 min (five times). Finally smooth muscle cell nuclei were stained with DAPI (1 ng/ml), and then images were obtained using imaging system 1 equipped with a 60× water dipping lens.

### Smooth muscle cell Ca$^{2+}$ imaging

All experiments used second-order mesenteric arteries. For Ca$^{2+}$ imaging experiments arteries were isolated, cut open and pinned out flat on a customized Sylgard silicon bath chamber, with smooth muscle cells facing up. Smooth muscle cells were then loaded with the fluorescent Ca$^{2+}$ indicator, Cal520/AM (5 µM, in DMSO with 0.025% Pluronic F-127), at 37 °C for 30 min. The myosin light-chain kinase inhibitor, wortmannin (10 µM), was included to prevent vasoconstriction during imaging. In experiments in which IP$_3$ was photoreleased to evoke a Ca$^{2+}$ response, the arteries were also loaded with a membrane-permeant photoactivatable form of IP$_3$ (cIP$_3$, 5 µM). Smooth muscle cell activity was recorded at room temperature (20°C) using either imaging system 1 or imaging system 2 for experiments with cIP$_3$.

A paired experimental approach (i.e. before and after in the same tissue) was used to investigate the levels of smooth muscle Ca$^{2+}$ activity under basal (unstimulated) conditions, depolarization-induced conditions (high K$^+$ PSS) and cIP$_3$-evoked conditions. Measurements were made before and after treatment with pharmacological agents as outlined in the text. In every experiment each replicate was obtained from a different animal, ensuring biological independence.

Basal and cIP$_3$-evoked activity was examined in the absence of flow, whereas high K$^+$-evoked activity was examined during superfusion of PSS containing 30 mM K$^+$ (high K$^+$ PSS; equimolar replacement for Na$^+$) at a flow rate of 1.5 ml/min (2-min recording, 30-s baseline). In all Ca$^{2+}$ experiments control responses were first obtained in PSS or in high K$^+$ PSS superfused into the bath at a flow rate of 1.5 ml/min. After each stimulation arteries were washed with PSS for 10 min and re-equilibrated for 10 min before subsequent recordings. During this time arteries were incubated with various pharmacological agents or subjected to pharmacological manipulations such as superfusion with Ca$^{2+}$-free PSS (with 1 mM EGTA, equimolar substitution with Mg$^+$) before Ca$^{2+}$ activity was recorded again. The equilibrium potential in high K$^+$ (30 mM) calculated using the Nernst equation was approximately –40 mV.

### Dual imaging of smooth muscle Ca$^{2+}$ and mitochondrial membrane potential ($\triangle\psi$)

For dual Ca$^{2+}$ and mitochondrial imaging experiments, arteries were cut open and pinned out flat on a customized Sylgard silicon bath chamber, with endothelial cells facing up. The endothelium was removed by gently scraping the intimal surface with a fine strand of hair. Smooth muscle cells were loaded with Cal-520/AM, as described earlier, and smooth muscle cell mitochondria were then stained with the membrane potential-sensitive dye TMRE (60 nM) for 5 min in PSS. Two-channel Cal-520/AM/TMRE fluorescence imaging was performed using a 60× water dipping lens at 10 Hz for 2 min using imaging system 3.

## ROS and superoxide detection

In experiments assessing the production or ROS or superoxide, the ROS-ID Total ROS/Superoxide Detection Kit from Enzo Life Sciences (ENZ-51010) was used. In brief the oxidative stress detection reagent and superoxide detection reagent were reconstituted in DMSO to yield a 5 mM stock. The endothelium was removed from the arteries, which were then incubated with either the oxidative stress detection reagent or the superoxide stress detection reagent (2 μM, 30 min) at 37°C. The samples were then incubated with either oligomycin (1 μM, 20 min) or PSS as control. Reagents were washed with PSS solution, and the arteries were stained with DAPI (1 ng/ml) before imaging. ROS (excitation wavelength: 490 nm) and superoxide production (excitation wavelength: 550 nm) were detected using imaging system 1. Separate preparations were incubated with detection reagents together with either a ROS inducer (pyocyanin, 500 μM), as a positive control, or a ROS inhibitor (*N*-acetyl-L-cysteine, 5 mM), as a negative control, for 30 min at 37°C.

## Data analysis

$Ca^{2+}$ signals in were measured in the same field of cells before (control) and after pharmacological manipulations. Subtle shifts in cell positions between each recording were corrected prior to $Ca^{2+}$ signal analysis. In brief circular regions of interest (ROIs) were generated for each cell (radius = 10 μm) and assigned a unique identification number so that each cell could be tracked throughout the recordings. ROI shifts between recordings were corrected using a customized Image J plugin.

$Ca^{2+}$ signals were extracted from each ROI and analysed using custom-written Python software (Wilson et al., 2016). Fluorescence signals were expressed as ratios ($F/F_0$) of fluorescence counts ($F$) relative to baseline values ($F_0$). Baseline values were calculated by averaging the fluorescence from the 100 frames where signals exhibited the least noise before stimulation.

Intracellular $Ca^{2+}$ signals evoked by plasma membrane depolarization consist of two components: a 'slow', persistent $Ca^{2+}$ elevation and fast $Ca^{2+}$ oscillations. These two components respond to various pharmacological interventions differently, indicating that distinct processes are involved. Slow and fast components were therefore analysed separately (Heathcote et al., 2019). Slow $Ca^{2+}$ elevations were extracted using an asymmetric least square (ALS) smoothing function. Fast $Ca^{2+}$ oscillations were isolated by normalizing each $F/F_0$ signal to the ALS-smoothed trace. Cells were classified as exhibiting slow $Ca^{2+}$ elevations when the extracted amplitude exceeded the background noise values by more than 10-fold. The amplitude of the slow signal was calculated by averaging the slow $F/F_0$ over the duration of the high $K^+$ PSS or agonist/antagonist exposure.

Cells were considered to exhibit fast $Ca^{2+}$ oscillations if they displayed at least one $Ca^{2+}$ spike that exceeded five times the standard deviation of baseline noise. All spikes were identified using a zero-crossing peak-detection algorithm (Wilson et al., 2016). The mean amplitude of the fast $Ca^{2+}$ oscillations was calculated by averaging the amplitude of all $Ca^{2+}$ spikes over the duration of high $K^+$ PSS or agonists/antagonists (Heathcote et al., 2019).

In mitochondrial imaging experiments ROIs corresponding to mitochondrial clusters were outlined using the 'Freehand ROI tool' in Image J (Schneider et al., 2012). The mean grey value of each ROI was measured and normalized to background fluorescence in Image J. The corrected mean fluorescence intensity for each artery was calculated by averaging the mean grey values of three different ROIs. In ROS/Superoxide imaging experiments, fluorescence intensity was measured from a ROI encompassing the entire field of view.

## Statistical analysis

All experiments followed a paired (within-subjects) design and are presented graphically as individual data points connected by lines. A single artery from each animal contributed one data series per experiment. For all experiments $n$ represents the number of biological replicates (i.e. number of animals). In each experiment measurements of $Ca^{2+}$ signalling properties (e.g. amplitude of fast and slow responses) are independent analyses conducted on the same number ($n$) of biological replicates. Data normality was assessed using the Shapiro–Wilk test. Single comparisons were analysed using a paired $t$ test if normality was confirmed. Multiple comparisons were analysed using repeated-measures one-way ANOVA with the Geisser–Greenhouse correction (to account for violations of sphericity) and Dunnet's multiple comparisons test (comparing each treatment to control). If normality was not met data were analysed using the Friedman test (non-parametric repeated-measures ANOVA) followed by a Dunn's multiple comparisons test. In one series of experiments basal $Ca^{2+}$ responses and stimulus-evoked $Ca^{2+}$ responses were assessed in the absence and presence of pharmacological agents in the same tissue. These data were assessed first for normality, and then using repeated-measures two-way ANOVA with the Geisser-Greenhouse correction and the uncorrected Fisher's least significant difference (LSD) test. In one case we examined the effects of an inhibitor in the absence and presence of another drug. These experiments were

unpaired and analysed using Welch's *t* test. All statistical tests were two sided, with $P < 0.05$ considered statistically significant. Specific statistical tests for each dataset are indicated in the respective figure legends. All statistical analysis was performed in GraphPad Prism 10.4.1.

## Results

To investigate voltage-dependent $Ca^{2+}$ entry in intact vascular smooth muscle cells, intracellular $Ca^{2+}$ signals were examined in second-order rat mesenteric arteries (Fig. 1A). Expression of L-type voltage-dependent

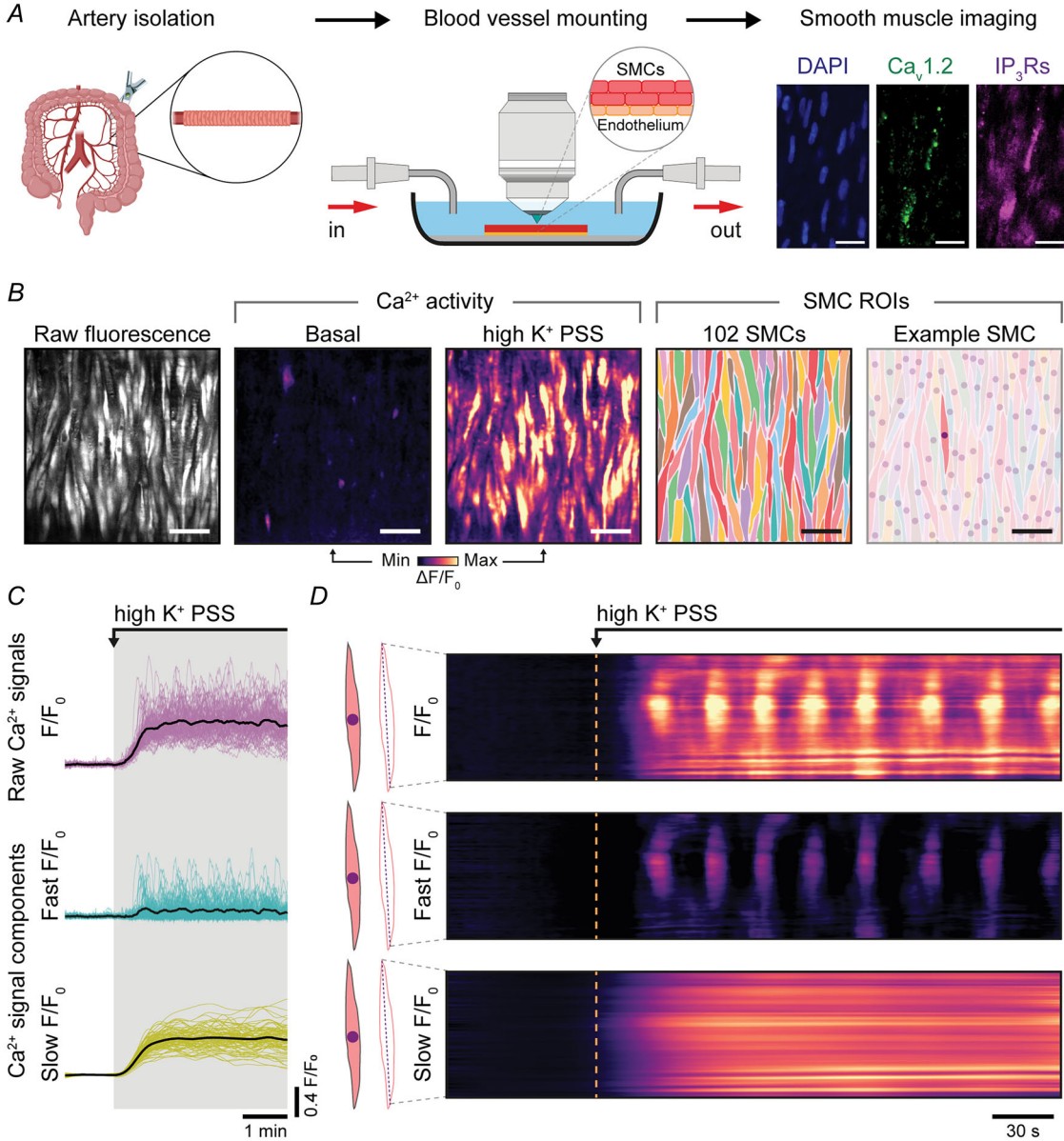

**Figure 1. Voltage-dependent $Ca^{2+}$ channels evoke repetitive $Ca^{2+}$ oscillations in intact smooth muscle cells**

*A*, schematic of the imaging system and artery preparation. Intact mesenteric smooth muscle *en face* preparations were isolated and loaded with fluorescent indicators. Right panel, $Ca_v1.2$ and $IP_3R$ (inositol triphosphate receptor) localization in smooth muscle cells. *B*, left to right, raw image of intact smooth muscle cells loaded with Cal-520; $Ca^{2+}$ heat map image illustrating oscillatory $F/F_0$ activity at rest (basal) and after high $K^+$ (30 mM) PSS (physiological salt solution); segmented smooth muscle cell images and their assigned regions of interest for $Ca^{2+}$ signalling analysis. Scale bar = 50 μm. *C*, top, overlaid $Ca^{2+}$ traces from all cells in (*B*) during high $K^+$ PSS perfusion (10 min; black lines represent the average). Fast $Ca^{2+}$ oscillations (middle) and slow, persistent $Ca^{2+}$ elevations (bottom) deconvoluted from the raw $Ca^{2+}$ signals. *D*, kymograph representation of the overall signal (top), fast $Ca^{2+}$ oscillations (middle) and slow, persistent $Ca^{2+}$ elevation (bottom) from the example cell annotated in (*B*) over the course of high $K^+$ PSS perfusion.

$Ca^{2+}$ channels (L-VDCCs, $Ca_V$ 1.2) in intact smooth muscle tissue was confirmed via immunohistochemistry (Fig. 1*A*). Depolarization of the plasma membrane with high $K^+$ PSS evoked an increase in cytoplasmic $Ca^{2+}$ concentration (Fig. 1*B*).

Two components of the $Ca^{2+}$ increase seemed to be evoked by voltage-dependent $Ca^{2+}$ entry (Fig. 1*C* and *D*). The first was a sustained slow $Ca^{2+}$ increase that was approximately uniform throughout the cytoplasm. The second was a series of periodic increases in $Ca^{2+}$, which propagated as spatial gradients of the ion through the cell ($Ca^{2+}$ waves; Fig. 1*C* and *D*). The slow, persistent increase is consistent with $Ca^{2+}$ entry via VDCCs. However because the smooth muscle plasma membrane is generally considered to be an isopotential surface (Singer & Walsh, 1980), propagating $Ca^{2+}$ waves are not consistent with direct voltage-dependent $Ca^{2+}$ entry to a sustained depolarization. Instead these waves are reminiscent of $Ca^{2+}$ release from the internal store. These results indicate that $Ca^{2+}$ entry may evoke $Ca^{2+}$ waves by triggering $Ca^{2+}$ release from the internal store.

The role of voltage-dependent $Ca^{2+}$ entry in the $Ca^{2+}$ increase evoked by high $K^+$ PSS was investigated by removing external $Ca^{2+}$ or by inhibiting VDCCs using the dihydropyridine blocker, nimodipine (10 μM; Fig. 2; $n = 5$ animals in each experiment). Both the slow, persistent $Ca^{2+}$ elevation and fast $Ca^{2+}$ oscillations were eliminated (Fig. 2*A*). These results suggest that $Ca^{2+}$ entry

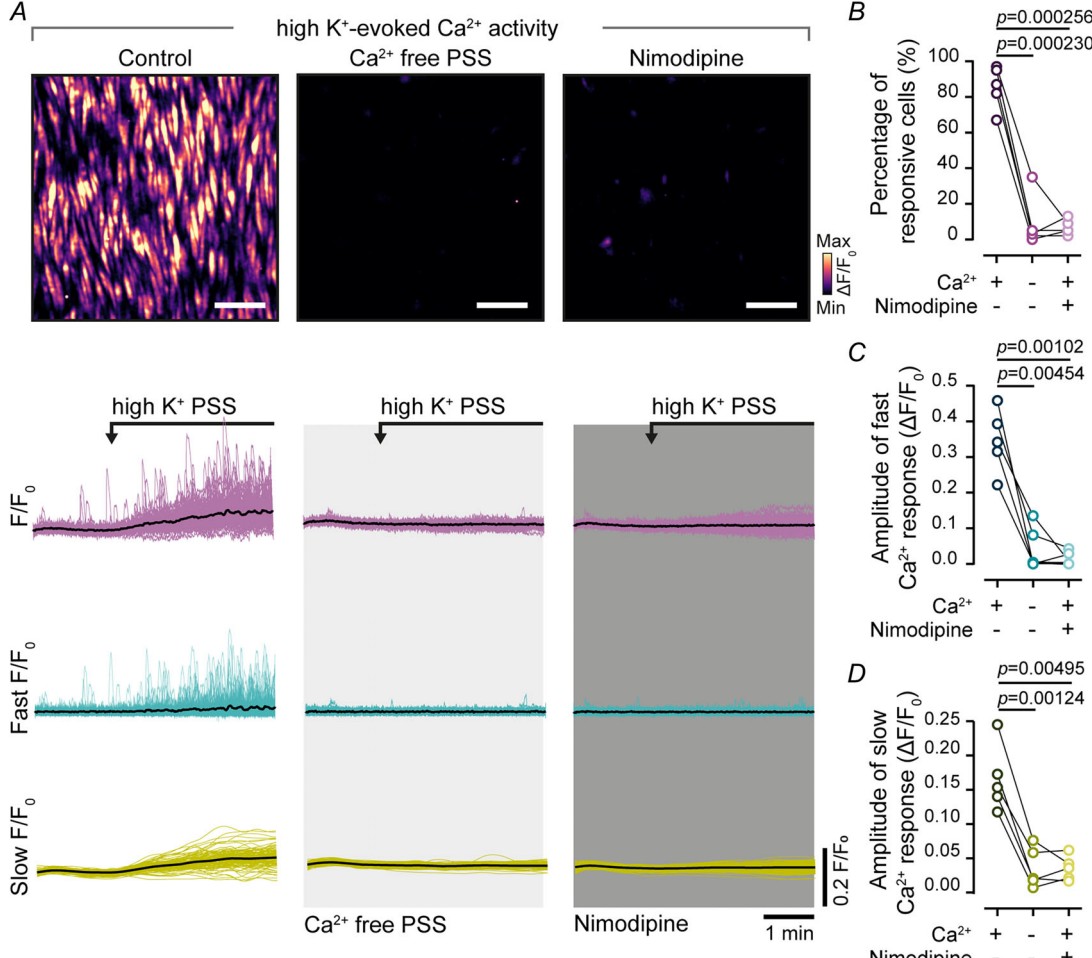

**Figure 2. VDCCs (voltage-dependent $Ca^{2+}$ channels) mediate $Ca^{2+}$ influx in intact smooth muscle cells**
*A*, heat map representations and $Ca^{2+}$ signals illustrating high $K^+$-evoked $Ca^{2+}$ activity under control conditions after the removal of extracellular $Ca^{2+}$ and after inhibition of L-type $Ca^{2+}$ channels using nimodipine (10 μM). Scale bar = 50 μm. Raw $Ca^{2+}$ signals were deconvolved into fast and slow $F/F_0$ signal components, and traces from all cells are shown overlaid. In each trace the bold black line is the averaged $Ca^{2+}$ signals from all cells. *B–D*, summarized data showing the percentage of cells activated by a high $K^+$ PSS (physiological salt solution), the mean amplitude of fast $Ca^{2+}$ responses (*C*) and the mean amplitude of slow, persistent $Ca^{2+}$ responses (*D*). Statistical significance ($P < 0.05$) using repeated-measures one-way ANOVA with the Geisser–Greenhouse correction and Dunnett's multiple comparisons test ($n = 5$ animals in each experiment *vs.* control).

via VDCCs is required to trigger both the slow, persistent $Ca^{2+}$ increase and $Ca^{2+}$ waves.

To investigate the mechanism by which $Ca^{2+}$ entry via VDCCs triggers repetitive $Ca^{2+}$ waves, the role of receptors involved in $Ca^{2+}$ release from the SR was examined. $Ca^{2+}$ release from the internal store may occur via $IP_3Rs$ or RyRs or both (Wray & Burdyga, 2010). $IP_3Rs$ and RyRs are each activated by cytosolic $Ca^{2+}$ (Hamada & Mikoshiba, 2012) leading to $Ca^{2+}$-induced $Ca^{2+}$ release (CICR) (Bosanac et al., 2002; Santulli & Marks, 2015). We aimed to identify the role of $IP_3Rs$ and RyRs in the $Ca^{2+}$ response triggered by VDCC-mediated $Ca^{2+}$ entry. $Ca^{2+}$ signals evoked by voltage-dependent $Ca^{2+}$ entry were examined before and after inhibition of $IP_3Rs$ with 2-APB (100 μM, 10 min) and inhibition of RyRs with dantrolene (10 μM, 10 min).

Blocking $IP_3Rs$ significantly decreased the amplitude of the high $K^+$ PSS-evoked $Ca^{2+}$ oscillations, whereas the slow, persistent $Ca^{2+}$ increase was unchanged (Fig. 3A; n = 5 animals). Subsequent blockade of RyRs had no further effect. In separate experiments initial inhibition of RyRs had no effect on the amplitude of the $Ca^{2+}$ oscillations or the amplitude of the slow, persistent $Ca^{2+}$ elevation (Fig. 3B; n = 5 animals). However the subsequent blockade of $IP_3Rs$ significantly reduced the amplitude of high $K^+$-evoked $Ca^{2+}$ oscillations (Fig. 3D).

Because $IP_3Rs$ and RyRs mediate $Ca^{2+}$ release from a common internal store (McCarron & Olson, 2008; Rainbow et al., 2009), we further investigated the source of repetitive $Ca^{2+}$ oscillations. First we depleted the internal store by opening RyRs with caffeine (10 mM) and ryanodine (50 μM), which locks RyRs in a persistently open subconductance state (Xu et al., 1994). Caffeine (10 mM) was repetitively, transiently applied in presence of ryanodine until the $Ca^{2+}$ increase ceased (not shown), confirming depletion of the store. After store depletion high $K^+$ PSS failed to evoke $Ca^{2+}$ oscillations, and the resulting signals consisted only of a slow, persistent $Ca^{2+}$ elevation (Fig. 3C; n = 5 animals).

Collectively these results suggest that $IP_3Rs$ and RyRs share access to the same internal $Ca^{2+}$ store and that $Ca^{2+}$ influx via VDCCs preferentially activates $IP_3Rs$ (rather than RyRs) to produce repetitive $Ca^{2+}$ oscillations and waves.

$Ca^{2+}$ release via $IP_3R$ after depolarization-evoked $Ca^{2+}$ influx is unlikely to result from activation of G proteins and phospholipase C, leading to $IP_3$ production and subsequently $IP_3$-mediated $Ca^{2+}$ release (del Valle-Rodríguez et al., 2003). In support the $Ca^{2+}$ increase evoked by depolarization of the plasma membrane with high $K^+$ PSS was unaltered by inhibition of phospholipase C with U73122 (2 μM; Fig. 4; n = 5 animals).

In addition to providing cellular energy, mitochondria act as a key store that modulates intracellular $Ca^{2+}$ signalling through $Ca^{2+}$ uptake and release. To investigate the role of mitochondria in regulating signals evoked by voltage-dependent $Ca^{2+}$ entry, the relationship between mitochondrial position, membrane potential and $Ca^{2+}$ signals was examined. Mitochondrial organization differs significantly in native *versus* cultured smooth muscle cells (McCarron, Wilson et al., 2013). In native smooth muscle mitochondria are organized in clusters (Chalmers et al., 2015, 2016) (Supplementary Video 1). To visualize mitochondria in smooth muscle cells, we used the mitochondrial membrane potential-sensitive dye TMRE (60 nM) alongside the $Ca^{2+}$ indicator Cal-520 (5 μM; Fig. 5A).

Mitochondria were predominantly organized in clusters (Fig. 5A) and remained immobile (Fig. 5B) when VDCCs were activated by high $K^+$ PSS. There was no obvious relationship between the position of the organelle and the origin of $Ca^{2+}$ waves. Furthermore neither mitochondrial position nor mitochondrial membrane potential changed during $Ca^{2+}$ waves (Fig. 5A and B). These results suggest that voltage-dependent $Ca^{2+}$ entry does not alter mitochondrial localization or membrane potential in intact smooth muscle cells.

Mitochondria may functionally interact with various $Ca^{2+}$-permeable channels to modulate channel gating and local/global $Ca^{2+}$ signals (Barstow et al., 2004; Chalmers & McCarron, 2008; Viola & Hool, 2010). Depolarization of the mitochondrial membrane is often used to prevent mitochondrial $Ca^{2+}$ uptake. Mitochondria in smooth muscle cells of intact arteries were rapidly depolarized by the oxidative phosphorylation uncoupler, CCCP (1 μM), but not by the ATP synthase inhibitor, oligomycin (1 μM; Fig. 5C and D; n = 5 animals).

To determine if mitochondria modulate signals generated by voltage-dependent $Ca^{2+}$ entry, high $K^+$-evoked $Ca^{2+}$ increases were examined before and after mitochondrial $Ca^{2+}$ uptake was prevented using CCCP. CCCP eliminated high $K^+$-evoked fast $Ca^{2+}$ oscillations without altering the slow, persistent $Ca^{2+}$ signals (Fig. 6A and C–E; n = 5 animals). The inhibition of $Ca^{2+}$ oscillations by CCCP was unaffected by the mitochondrially targeted antioxidant, MitoTEMPO (Fig. 6B and C–E; n = 5 animals). These results suggest that a polarized mitochondrial membrane is required for sustaining fast $Ca^{2+}$ oscillations and waves triggered by $Ca^{2+}$ entry via VDCCs in intact vascular smooth muscle.

ATP depletion may also contribute to the inhibition of the response by CCCP. To assess this possibility the effects of the ATP synthase inhibitor oligomycin was examined (Drumm et al., 2018). Oligomycin (1 μM) significantly inhibited the amplitude and frequency of high $K^+$ PSS evoked $Ca^{2+}$ oscillations (Fig. 7A, D and E; n = 5 animals). Once again the slow, persistent $Ca^{2+}$ elevation was unchanged (Fig. 7F).

However the inhibition of $Ca^{2+}$ oscillations appeared to result from ROS production triggered by oligomycin

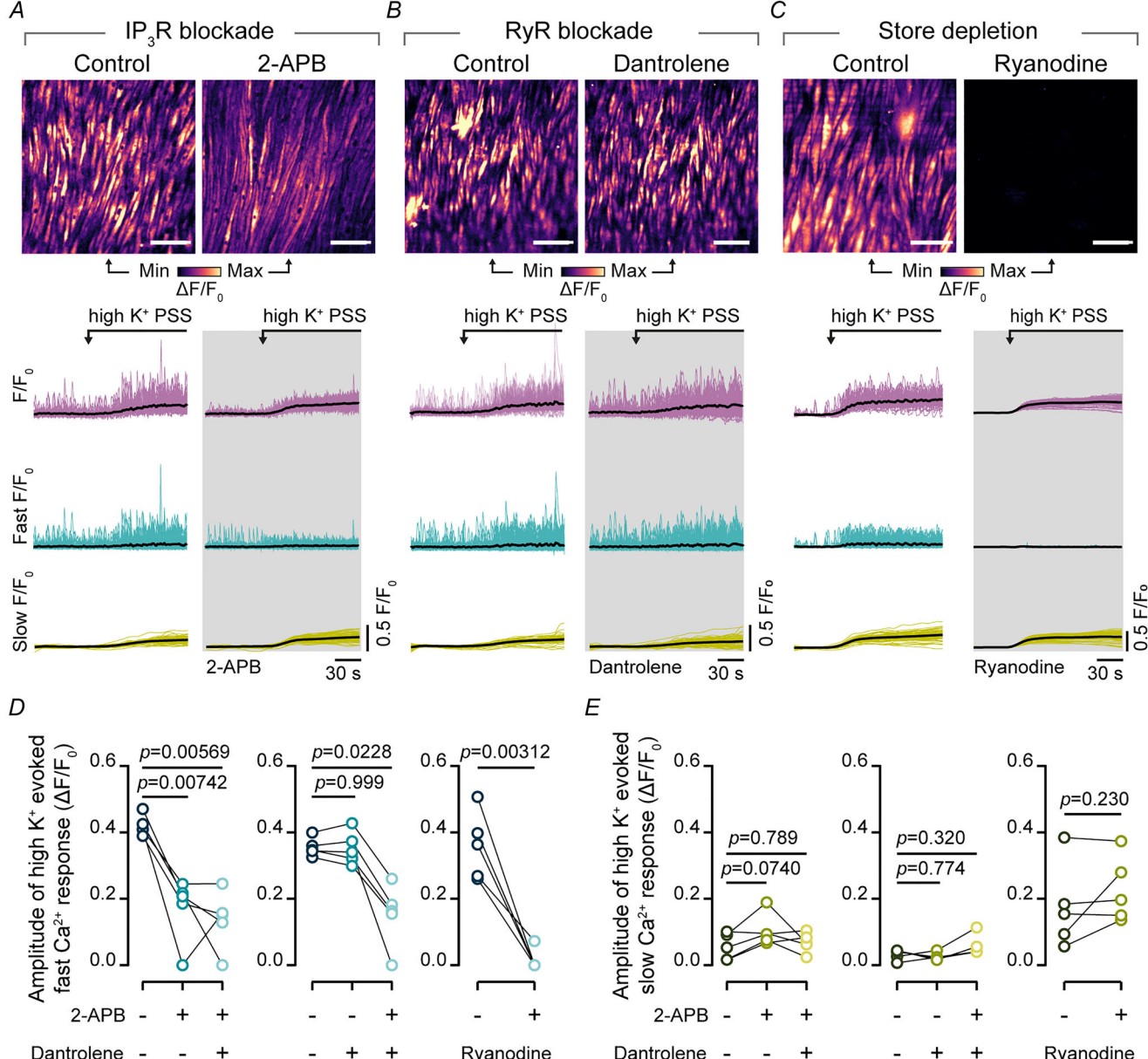

**Figure 3. IP$_3$R (inositol triphosphate receptor), not RyRs (ryanodine receptors), mediate the coupling of Ca$^{2+}$ influx to Ca$^{2+}$ oscillations and waves**

Data from three separate (*A–C*) experiments in which arteries were sequentially superfused with high K$^+$ PSS (physiological salt solution) before and after incubation with 2-APB (2-aminoethyl diphenylborinate, 100 μM) (*A*), dantrolene (10 μM) (*B*) or store depletion after repetitive intermittent caffeine (10 mM) applications in presence of ryanodine (50 μM) (*C*). There was a 10 min wash and 10 min rest between each recording. *A–C*, heat map representations (top) and Ca$^{2+}$ signals (bottom) illustrating high K$^+$-evoked Ca$^{2+}$ activity, scale bar = 50 μm. Raw Ca$^{2+}$ signals were deconvolved into fast and slow *F/F$_0$* signal components, and traces from all cells are shown overlaid. *D* and *E*, summarized data showing the mean amplitude of fast Ca$^{2+}$ oscillations (*D*) and the mean amplitude of slow, persistent Ca$^{2+}$ elevations (*E*). Statistical significance (*P* < 0.05) determined using repeated-measures one-way ANOVA with the Geisser–Greenhouse correction and Dunnett's multiple comparisons test or the Friedman test followed by Dunn's multiple comparisons test for 2-APB/dantrolene experiments, or paired *t* test for the experiments with ryanodine (*n* = 5 animals in each experiment).

rather than just by ATP depletion. Indeed oligomycin caused an increase in ROS but not superoxide levels in smooth muscle cells (Fig. 7C; $n = 6$ animals in each experiment). Furthermore the inhibitory effects of oligomycin were reduced by the mitochondrial-targeted antioxidant MitoTEMPO (10 μM; Fig. 7B and G; $n = 5$ animals). These experiments suggest that mitochondria regulate IP$_3$-evoked Ca²⁺ signals by at least two mechanisms. The first mechanism requires an intact membrane potential, whereas the second relies on the production of ROS by the organelle.

To determine whether or not IP$_3$ production or IP$_3$Rs activity is inhibited when mitochondrial function is compromised, IP$_3$Rs were directly activated by photolysis of cIP$_3$. This approach bypasses the production of the inositide. IP$_3$ was released in pre-selected areas (10,000 μm²) of the smooth muscle cells by flash photolysis of the caged inositide (flash duration: 1 ms). Arteries were then incubated with either oligomycin (1 μM) or CCCP (1 μM) and cIP$_3$ photoreleased again.

Photolysis of cIP$_3$ triggered repetitive Ca²⁺ spikes in the uncaging area, which propagated to neighbouring cells. Inhibition of ATP production with oligomycin (1 μM) had no effect on IP$_3$-evoked Ca²⁺ release. However subsequent depolarization of the mitochondrial membrane

with CCCP (1 μM) significantly reduced the amplitude of the Ca²⁺ response to photoreleased IP$_3$ and blocked Ca²⁺ propagation to neighbouring cells.

Collectively these results suggest that mitochondrial membrane potential is essential for depolarization-evoked IP$_3$R-mediated Ca²⁺ release and Ca²⁺ wave propagation in smooth muscle cells. Alterations in ROS and ATP production may modulate IP$_3$ production rather than IP$_3$R activation directly. Fig. 8.

## Discussion

Ca²⁺ entry via dihydropyridine-sensitive Ca²⁺ channels significantly influences the overall Ca²⁺ levels in smooth muscle (Berridge, 2008; Sanders, 2001; Sanders et al., 2024). In this study conducted on intact resistance arteries, we found that activation of VDCCs leads to Ca²⁺ entry, which in turn triggers Ca²⁺ release from the internal store via IP$_3$Rs, producing repetitive Ca²⁺ oscillations and waves. Notably the Ca²⁺ increases resulting from this release, but not from voltage-dependent Ca²⁺ entry, are significantly influenced by mitochondrial activity. These findings indicate that the Ca²⁺ signal from voltage-dependent Ca²⁺ entry has two elements, only one of which is modulated by mitochondria (Fig. 9).

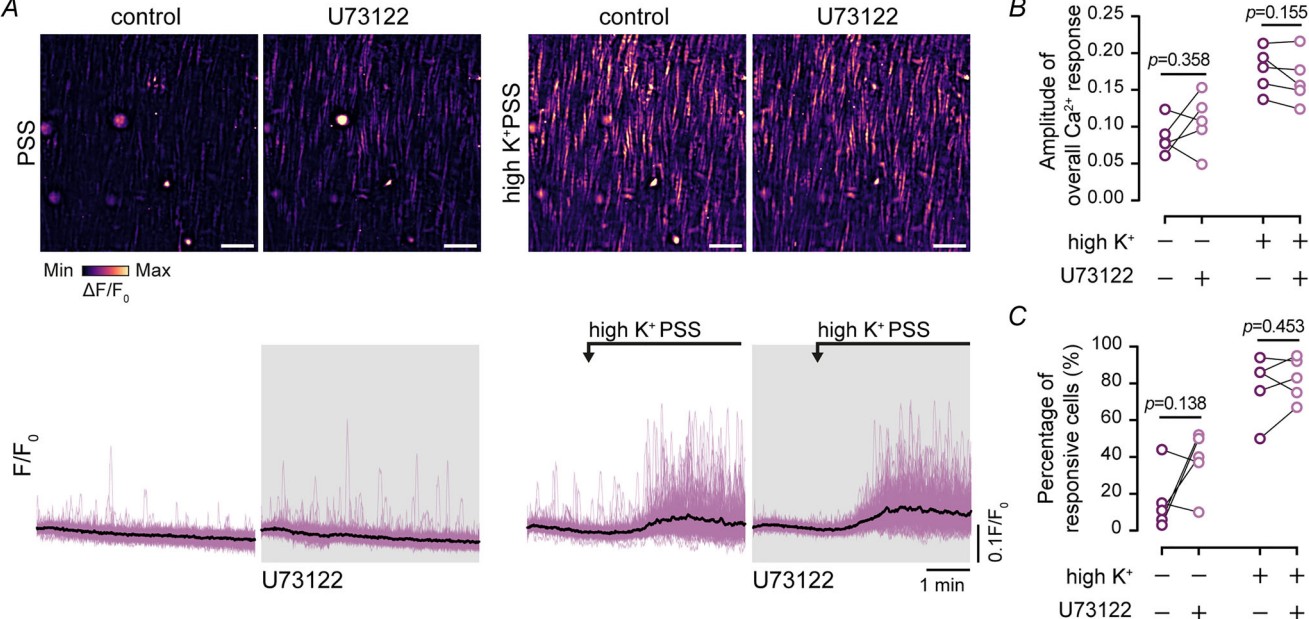

**Figure 4. IP$_3$ (inositol triphosphate) production is not required for high K⁺ PSS (physiological salt solution)-evoked Ca²⁺ oscillations**
*A*, top, heat map representation of basal (left) and high K⁺-evoked (right) Ca²⁺ activity in the absence and presence of the phospholipase C blocker, U73122 (2 μM). Bottom, raw Ca²⁺ signals from each cell shown in the heat map images. *B* and *C*, summarized data showing the mean amplitude (*B*) of high K⁺ PSS-evoked Ca²⁺ signals and the percentage of cells responding to high K⁺ PSS (*C*). Statistical significance was determined using a repeated-measured two-way ANOVA with the Geisser–Greenhouse correction and uncorrected Fisher's LSD (least significant difference) for multiple comparisons (treatment *vs.* control for basal activity, treatment *vs.* control for high K⁺-evoked activity; $n = 5$ animals).

$Ca^{2+}$ influx during voltage-dependent $Ca^{2+}$ entry generates a uniform and sustained increase in bulk average $Ca^{2+}$ concentration determined by the total $Ca^{2+}$ current delivered in individual smooth muscle cells (Kamishima & McCarron, 1996). However in intact blood vessels where contact between cells is maintained, voltage-dependent $Ca^{2+}$ entry induced by high $K^+$ PSS causes repetitive $Ca^{2+}$ oscillations and waves across the entirety of the smooth muscle cells in the artery. Because the membrane potential is largely clamped by high $K^+$ PSS, these $Ca^{2+}$ oscillations indicate a complexity to the $Ca^{2+}$ signal that goes beyond ion entry when VDCCs are activated.

We identified two distinct components of the $Ca^{2+}$ changes triggered by voltage-dependent $Ca^{2+}$ entry. The first is a slow, persistent and steady increase in $Ca^{2+}$, which is expected from voltage-dependent $Ca^{2+}$ entry. The second component, characterized by rapid, repetitive oscillations with distinct spikes and propagating $Ca^{2+}$ waves, is also triggered by $Ca^{2+}$ entry via VDCCs but requires an additional mechanism, that is, $Ca^{2+}$ release from the internal store. Supporting the role of $Ca^{2+}$ release after $Ca^{2+}$ entry, when internal stores are depleted of $Ca^{2+}$, high $K^+$ PSS generated only a slow, steady increase in intracellular $Ca^{2+}$, which results only from voltage-dependent $Ca^{2+}$ entry.

Repetitive $Ca^{2+}$ oscillations and waves are a common feature of vascular smooth muscle cells to $IP_3$-generating agonists, such as uridine triphosphate (UTP; Jaggar et al., 2000), noradrenaline (norepinephrine) (Boittin et al., 1999; Iino et al., 1994; Miriel et al., 1999; Ruehlmann et al., 2000) and acetylcholine (Sevilla et al., 2008). These oscillations result from $IP_3$-evoked $Ca^{2+}$ release from internal stores. This $Ca^{2+}$ release is a self-reinforcing process which, once initiated, produces a rapid increase

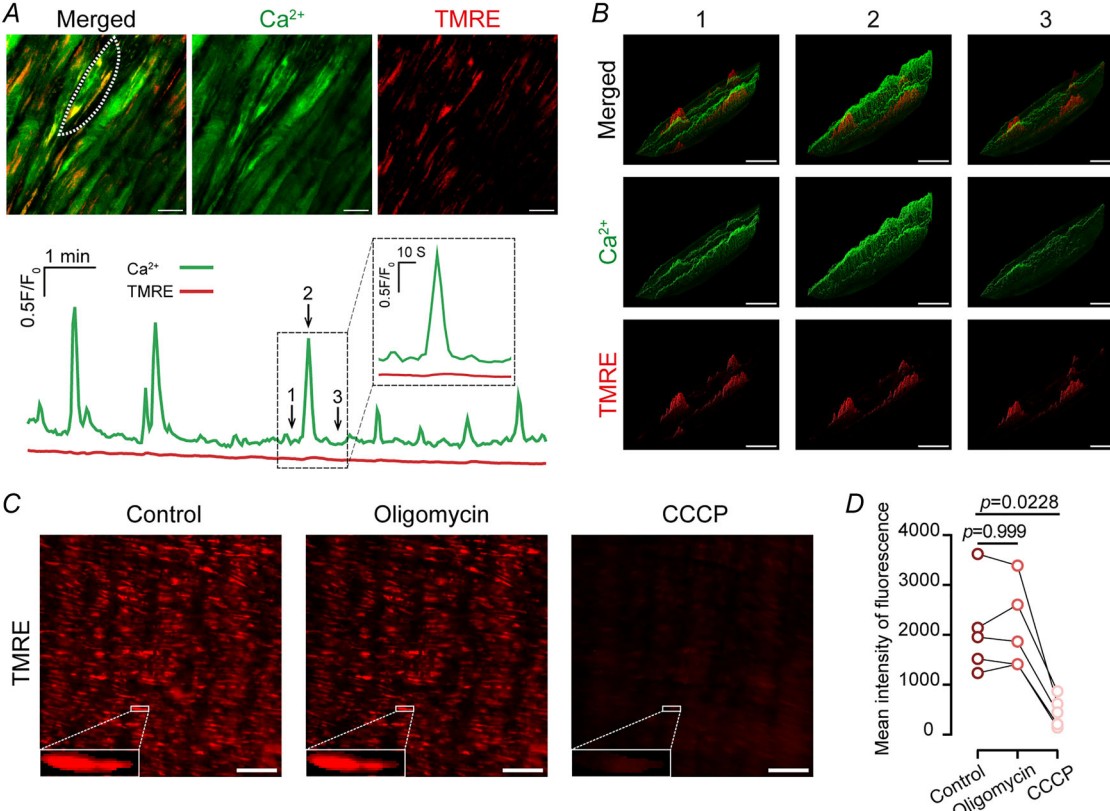

**Figure 5. Mitochondria are immobile and do not depolarize during or after large $Ca^{2+}$ increases in intact mesenteric artery smooth muscle cells**

*A*, top, representative images showing intracellular $Ca^{2+}$ (Cal520, green) and mitochondrial membrane ($\Delta\Psi m$, TMRE (tetramethylrhodamine ethyl ester perchlorate), red) in intact smooth muscle cells, scale bar = 10 μm. Bottom, traces of $Ca^{2+}$ signals (green) and TMRE fluorescence (red) from the region indicated by the white dashed lines in the top panel. $Ca^{2+}$ activity was stimulated with high $K^+$ PSS. *B*, 3-D surface plots of merged channels (top), $Ca^{2+}$ signals (middle) and $\Delta\Psi m$ (bottom) at the time points indicated in *A* (bottom). These plots illustrate the intensity and localization of intracellular $Ca^{2+}$ and mitochondrial membrane potential during a $Ca^{2+}$ event in the indicated area in (*A*), scale bar = 10 μm. *C*, representative images showing the effects of oligomycin (1 μM) and CCCP (*m*-chlorophenylhydrazone, 1 μM) on smooth muscle mitochondrial membrane potential. Scale bar = 50 μm. *D*, summary data of TMRE fluorescence intensity across treatment groups. Statistical significance ($P < 0.05$) was determined using Friedman test followed by Dunn's multiple comparisons test (*vs*. control, *n* = 5 animals).

in $Ca^{2+}$ levels that terminates due to local depletion of the store or deactivation of the receptors responsible for $Ca^{2+}$ release (McCarron et al., 2007). Our results indicate that this process is also activated during depolarization-evoked $Ca^{2+}$ entry.

Support for the involvement of $IP_3$ receptors in the process is found in the observation that $IP_3R$ inhibitors significantly reduce $Ca^{2+}$ oscillations induced by high $K^+$ PSS, whereas inhibiting RyRs does not. Furthermore because $IP_3Rs$ and RyRs share access to a common $Ca^{2+}$ pool in smooth muscle (McCarron & Olson, 2008; Rainbow et al., 2009), depleting the store via RyRs should

reduce the response to $IP_3$ activation. Indeed we observed that depleting the internal store via RyRs inhibited the oscillations. On the contrary neither $IP_3Rs$ nor RyRs contribute to the steady, persistent increase in $Ca^{2+}$ triggered by high $K^+$ PSS. The additional $Ca^{2+}$ release from the internal store via $IP_3Rs$ provides a means to activate additional physiological activities that require transient changes in cytoplasmic $Ca^{2+}$ concentration during voltage-dependent $Ca^{2+}$ entry.

The mechanism by which $Ca^{2+}$ entry via VDCCs triggers $IP_3$-evoked $Ca^{2+}$ release is unclear. One possibility is a CICR-like process, where $Ca^{2+}$ entry

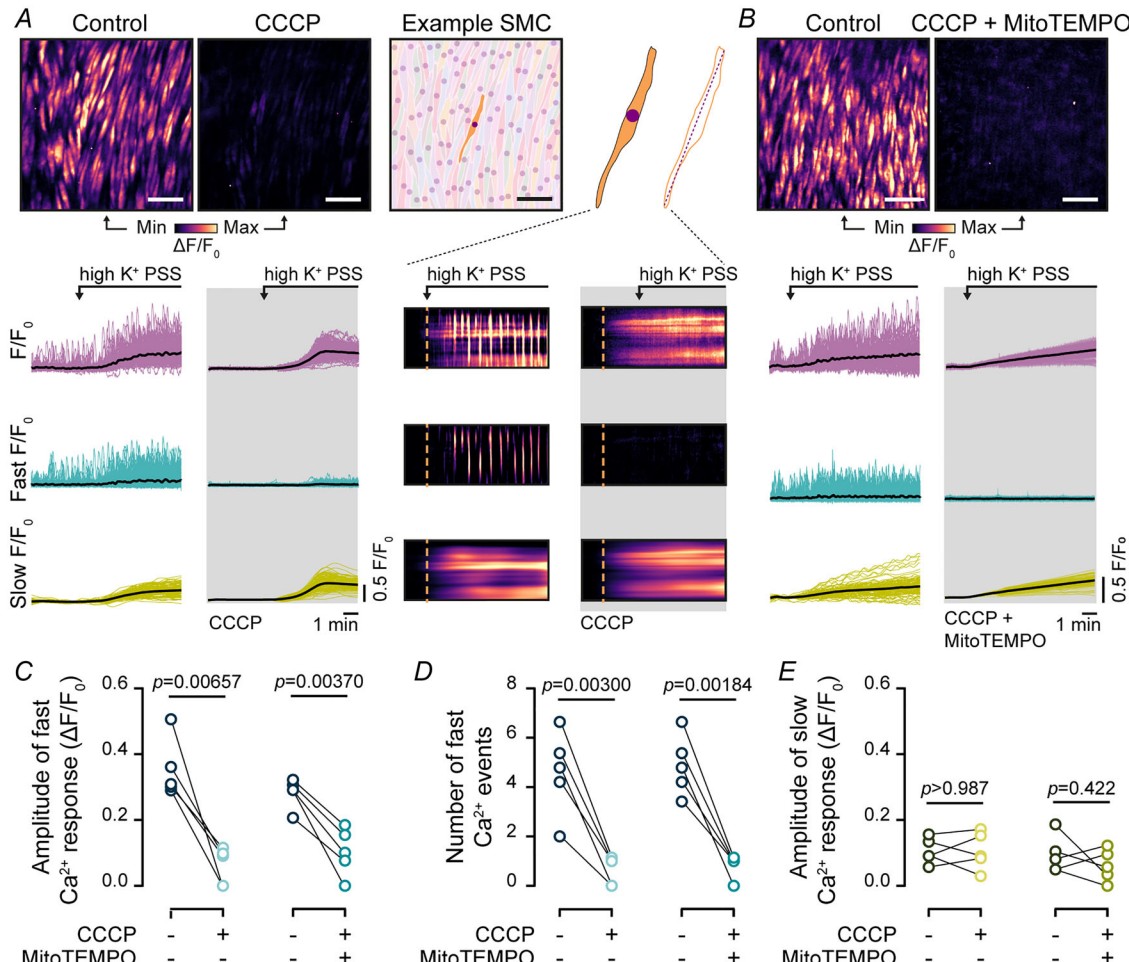

**Figure 6. Mitochondria regulate $Ca^{2+}$ oscillations evoked by VDCC (voltage-dependent $Ca^{2+}$ channel) activity**

*A* and *B*, heat map representations and $Ca^{2+}$ signals illustrating high $K^+$-evoked $Ca^{2+}$ activity in the absence and presence of the mitochondrial oxidative phosphorylation uncoupler (CCCP (*m*-chlorophenylhydrazone)), 1 μM (*A*) or with CCCP and a mitochondrially targeted antioxidant (MitoTEMPO), 10 μM (*B*). Scale bar = 50 μm. Raw $Ca^{2+}$ signals were deconvolved into fast and slow *F/F₀* signal components, and traces from all cells are shown overlaid. The bold black line is the averaged signal from all cells. In (*A*), a segmented image highlighting individual smooth muscle cells is shown, and kymographs show the $Ca^{2+}$ signal components from the indicated example cell (*A*, right). *C–E*, summary data showing the mean amplitude of fast $Ca^{2+}$ oscillations (*C*), the number of $Ca^{2+}$ oscillations (*D*) (measured over the entire analysis period of 10 min) and the amplitude of the slow $Ca^{2+}$ elevation (*E*) evoked by high $K^+$ PSS. Statistical significance ($P < 0.05$) determined using paired *t* test ($n = 5$ animals in each experiment).

via VDCCs activates IP₃Rs. IP₃Rs are regulated by $Ca^{2+}$-dependent feedback mechanisms and may be recruited by increased $Ca^{2+}$ levels. The production of $Ca^{2+}$ oscillations and waves may depend on these $Ca^{2+}$-dependent feedback processes (McCarron et al., 2006). After an initial influx of $Ca^{2+}$ through VDCCs, IP₃Rs might facilitate a CICR-like process, triggering further $Ca^{2+}$ release from stores in a regenerative positive

feedback loop. IP₃Rs are more sensitive to activation by $Ca^{2+}$ when compared to RyRs (except in cases of 'store overload', reviewed: McCarron et al., 2006). This sensitivity likely underlies the primary role of IP₃R in facilitating $Ca^{2+}$ release. Supporting this proposal depolarization did not cause repetitive oscillations when VDCCs were blocked or when extracellular $Ca^{2+}$ was absent. Thus without $Ca^{2+}$ influx neither VDCC activity

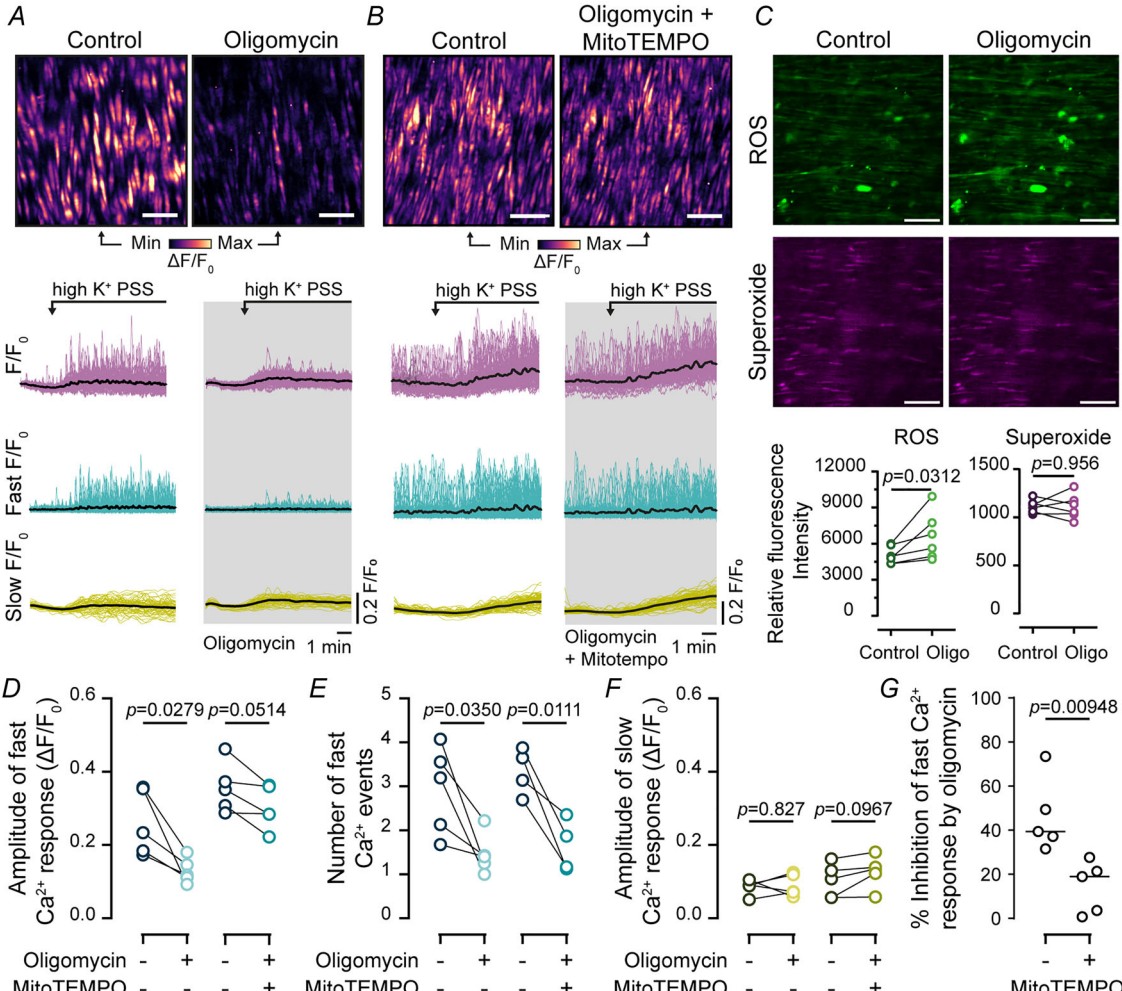

**Figure 7. Inhibition of ATP production modulates $Ca^{2+}$ oscillations evoked by VDCCs (voltage-dependent $Ca^{2+}$ channels)**
*A* and *B*, heat map representations and $Ca^{2+}$ signals showing high K⁺-evoked $Ca^{2+}$ activity in the absence and then presence of the ATP synthase inhibitor (oligomycin, 1 μM) (*A*) and oligomycin (1 μM) together with a mitochondrial-targeted antioxidant (MitoTEMPO, 10 μM) (*B*). Scale bar = 50 μm. Raw $Ca^{2+}$ signals were deconvolved into fast and slow *F/F₀* signal components, and traces from all cells are shown overlaid. The bold black line is the averaged $Ca^{2+}$ signals from all cells (*A, B*, bottom). *C*, images of smooth muscle ROS (reactive oxygen species) and superoxide levels in the absence or presence of oligomycin (top). Summary data show mean fluorescence intensity across the field of view (14,231 μm²) (bottom). *D–F*, summary data showing the mean amplitude of fast $Ca^{2+}$ oscillations (*D*), the number of fast $Ca^{2+}$ oscillations (*E*) (measured over the entire analysis period of 10 min) and the amplitude of the slow $Ca^{2+}$ elevations (*F*) evoked by high K⁺ PSS (physiological salt solution). *G*, summary data showing the inhibitory effects of oligomycin in the absence and presence of MitoTEMPO. Statistical significance ($P < 0.05$) determined using paired *t* test (superoxide data) or the Wilcoxon matched-pairs signed rank test (ROS (reactive oxidative species); $n = 6$ in each experiment). For $Ca^{2+}$ imaging data statistical significance ($P < 0.05$) was determined using paired *t* tests ($n = 5$ animals in each experiment). Welch's *t* test was used to assess the inhibition data in panel *G*.

nor changes in membrane potential significantly alter $Ca^{2+}$ levels. Therefore $Ca^{2+}$ oscillations in response to plasma membrane depolarization are secondary to $Ca^{2+}$ influx via VDCCs.

The current results are unlikely to be explained by depolarization-induced activation of G proteins and phospholipase C, leading to $IP_3$ production and subsequently $IP_3$-mediated $Ca^{2+}$ release (del Valle-Rodríguez et al., 2003). In contrast to the present findings, depolarization-induced activation of G proteins and phospholipase C does not require $Ca^{2+}$ entry. Furthermore in the present study depolarization-evoked $IP_3$-mediated $Ca^{2+}$ release was unaltered by inhibition of phospholipase C.

Another possibility, in line with the changes in activity that occur at RyRs on the internal store, is that $Ca^{2+}$ uptake during depolarization-evoked $Ca^{2+}$ entry may result in 'store overload'. Increases in $Ca^{2+}$ within the SR lumen, after elevation in bulk average $Ca^{2+}$ concentration, which results in store overload, are accompanied by an increase in RyR activity (McCarron et al., 2006). This mechanism may contribute to the generation of CICR at RyRs after $Ca^{2+}$ influx and to $Ca^{2+}$ release via RyR in $Ca^{2+}$ waves that had been initiated by $IP_3$ (McCarron et al., 2006). This seems unlikely in the present study as $IP_3$Rs do not

have a comparable luminal regulation as occurs at RyRs (Bezprozvanny & Ehrlich, 1994). Overall the most likely explanation of the results presented is that $Ca^{2+}$ entering the cell triggers a CICR-like process at $IP_3$Rs.

In smooth muscle the SR forms a continuous inter-connected network (McCarron & Olson, 2008; Rainbow et al., 2009) that often comes into close proximity with mitochondria. Indeed in mesenteric artery smooth muscle cells, mitochondria are positioned at a distance from the plasma membrane (Firth et al., 2009). At sites of close proximity with the SR, mitochondria are exposed to local concentrations of $Ca^{2+}$ that exceed average levels in the cell, potentially allowing the organelle to exert greater influence over the $Ca^{2+}$ signal. Indeed inhibiting mitochondrial function reduces $Ca^{2+}$ release through $IP_3$Rs. Two distinct mechanisms contribute to this inhibition, shedding light on how the organelles control $IP_3$Rs. The first is that polarized mitochondria are necessary to drive the uptake of $Ca^{2+}$ released from internal stores. Disrupting the mitochondrial membrane potential hampers this process, leading to a build-up of $Ca^{2+}$ near $IP_3$R, which inhibits further $Ca^{2+}$ release (Bezprozvanny et al., 1991; Szado et al., 2003). Consequently (paradoxically) by taking up $Ca^{2+}$, mitochondria decrease the local cytoplasmic $Ca^{2+}$

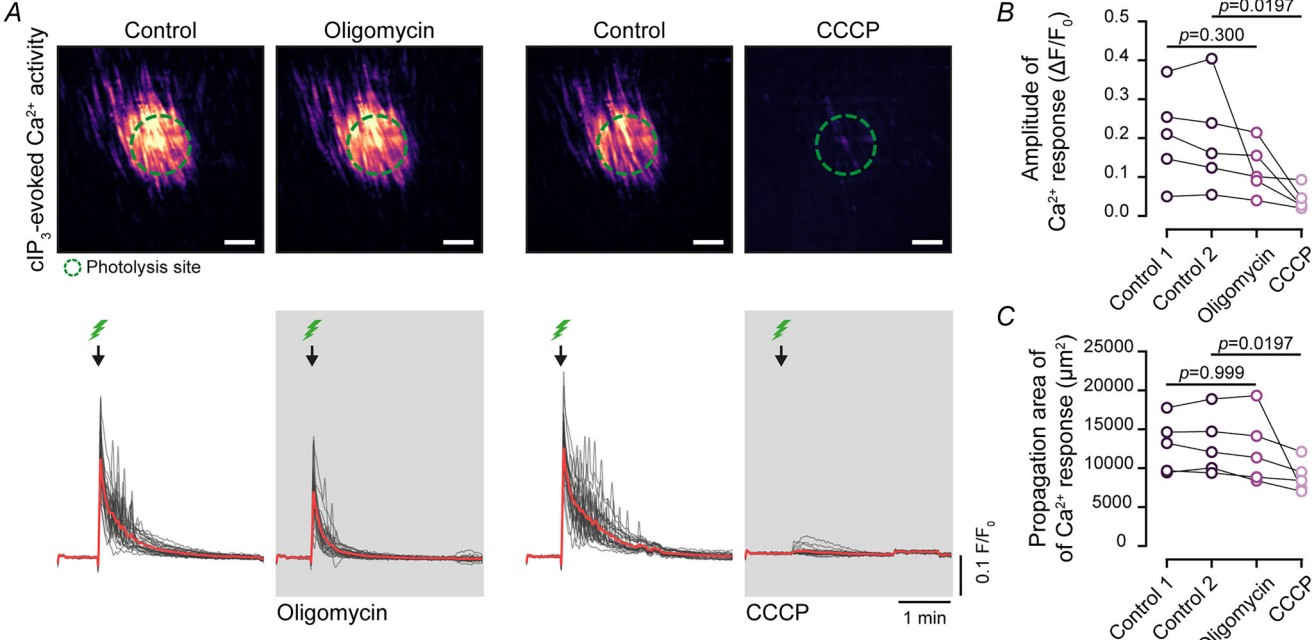

**Figure 8. Depolarization of the mitochondrial membrane potential suppresses $Ca^{2+}$ release from $IP_3$Rs (inositol triphosphate receptors) in smooth muscle cells**

*A*, top, heat map images of $Ca^{2+}$ signals in response to photolysis of caged $IP_3$ in arteries that have been denuded of endothelial cells. Dashed green circle: area of photolysis. Scale bar = 50 μm. Bottom, $Ca^{2+}$ traces from the single cells activated in the top panel. The bold red line is the averaged $Ca^{2+}$ signals from cells in the uncaging area. *B*, summary data of average amplitude of $Ca^{2+}$ peaks; (*C*) propagation area of $Ca^{2+}$ signals after photolysis. Statistical significance ($P < 0.05$) determined using the Friedman test with Dunn's multiple comparisons test ($n$ = 5 animals).

concentration, thus extending the duration of release from $IP_3Rs$ and increasing the overall cytoplasmic $Ca^{2+}$ concentration.

At locations near the SR, where mitochondria are exposed to high local concentrations of $Ca^{2+}$, depolarization of the mitochondrial membrane might be expected to occur. This depolarization may occur because of the larger $Ca^{2+}$ current produced during mitochondrial uptake of the ion triggered by the local $Ca^{2+}$ transients. For this process to occur mitochondria need to be anchored in position near the SR. However in certain cell types, such as proliferative smooth muscle cells, mitochondria are highly mobile and are less likely to be selectively positioned with the SR. Our findings demonstrate that in native vascular smooth muscle cells, mitochondria are predominantly immobile organelles. Furthermore we observed that the mitochondrial membrane potential remains unchanged during repetitive $Ca^{2+}$ signals. This suggests that despite the accumulation of $Ca^{2+}$ during $IP_3$-evoked $Ca^{2+}$ release, the membrane potential of mitochondria is unaffected.

Two explanations may account for the absence of a measured mitochondrial depolarization. The current carried by $Ca^{2+}$ may be insufficient to alter the inner mitochondrial membrane potential, or there may be a compensatory efflux of positively charged species (e.g. protons) counteracting the depolarization induced by $Ca^{2+}$ influx. Whereas depolarization has been reported in some studies (Yamamura, 2024; Yamamura et al., 2018), others, as in the present study, failed to detect significant mitochondrial membrane potential depolarization in response to transient increases in cytoplasmic $Ca^{2+}$ concentrations induced by $IP_3$-generating agonists (Chalmers & McCarron, 2008; Collins et al., 2000; Hajnóczky et al., 2003).

A second mechanism by which mitochondria may regulate $IP_3$-evoked $Ca^{2+}$ release appears to be largely independent of the mitochondrial membrane potential. When ATP production is inhibited by an ATP synthase inhibitor (oligomycin), there is a ROS production burst that inhibits $IP_3Rs$. In support of this conclusion the block of $IP_3$-evoked $Ca^{2+}$ release by the ATPase inhibitor is rescued by a mitochondrial-targeted antioxidant (MitoTEMPO). Interestingly although oligomycin significantly suppresses $Ca^{2+}$ release from $IP_3Rs$ evoked by voltage-dependent $Ca^{2+}$ entry, the inhibitor had a minimal effect on $Ca^{2+}$ release evoked by direct activation of $IP_3R$. These results suggest that ROS production may desensitize $IP_3Rs$ to $Ca^{2+}$ rather than altering the response to $IP_3$ itself. These findings confirm involvement of ROS production in the regulation of $IP_3$-evoked $Ca^{2+}$ release.

The role of ROS in $Ca^{2+}$ signalling is complex. In certain studies ROS produced by mitochondria might sensitize RyRs and $IP_3Rs$, leading to increased $Ca^{2+}$ release from the stores. However with increased concentrations, or more prolonged exposure to ROS,

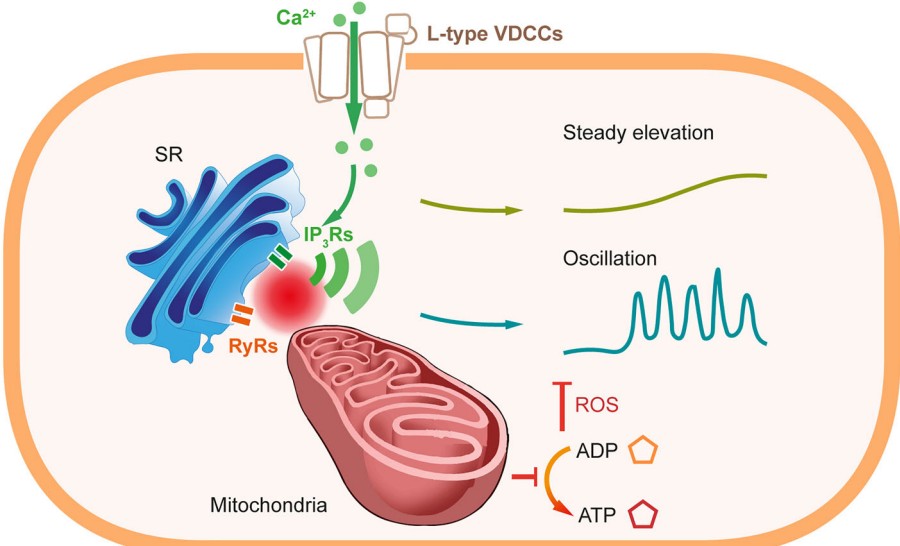

**Figure 9. Proposed mechanism of mitochondrial regulation of $Ca^{2+}$ signals resulting from VDCC activity in smooth muscle cells**

Depolarization of the plasma membrane induces $Ca^{2+}$ influx via L-type VDCCs that produces a steady elevation in $Ca^{2+}$, which activates $IP_3Rs$ to evoke repetitive $Ca^{2+}$ release events from the intracellular $Ca^{2+}$ store. Depolarization of the mitochondrial membrane potential reduces $IP_3R$ activity. Similarly inhibition of the ATP synthase increases ROS production, which also decreases $IP_3R$ activity. $Ca^{2+}$ entry via VDCC is unaffected by changes in the mitochondrial membrane potential or inhibition of the ATP synthase. $IP_3Rs$, inositol phosphate receptors; ROS, reactive oxygen species; RyRs, ryanodine receptors; SR, sarcoplasmic reticulum; VDCC, voltage-dependent $Ca^{2+}$ channel.

Ca$^{2+}$ release is suppressed (Yan et al., 2008). In native endothelial cells H$_2$O$_2$ inhibits IP$_3$Rs-mediated Ca$^{2+}$ release (Zhang et al., 2019). Superoxide typically lasts for very brief periods in the cytoplasm (seconds) before being converted into H$_2$O$_2$ by superoxide dismutase. ROS species like H$_2$O$_2$ can persist for longer periods. Therefore mitochondria may regulate Ca$^{2+}$ release from IP$_3$Rs under physiological conditions without the membrane potential collapsing, which typically occurs only under extreme conditions.

Studies examining the role of mitochondria in regulating Ca$^{2+}$ signalling often use pharmacological agents like protonophores to alter mitochondrial function. However as Drumm et al. (2018) highlight, protonophores can have effects that extend beyond the mitochondria, at least under certain conditions. These off-target effects include the indirect inhibition of the sodium pump due to ATP depletion (Tretter et al., 1998), plasma membrane depolarization through activation of H$^+$ and Na$^+$ currents (Park et al., 2002), changes in intracellular pH and the generation of free radicals (McCarron, Olson et al., 2013). Each of these effects will lead to steady-state changes in cell function.

In our study the protonophore CCCP inhibited repetitive Ca$^{2+}$ oscillations but did not alter the steady-state (slow) response. This suggests that off-target effects are unlikely to contribute to our findings. In support some of the reported off-target effects, such as those on the sodium pump, stem from the specific action of protonophores on ATP synthesis inhibition by collapse of the mitochondrial membrane potential. In our experiments inhibition of ATP synthesis by oligomycin (in presence of antioxidants) had no effect on oscillations, and the effect of CCCP was unchanged by antioxidants. This further supports the conclusion that off-target effects of protonophores are not likely to be responsible for our findings.

The widely used IP$_3$ receptor blocker 2-APB is also reported to exhibit off-target effects. These include activating some TRP channels or inhibiting Orai, K$^+$ channels and mitochondrial Ca$^{2+}$ uptake. None of these off-target effects explain the present findings. In the present study 2-APB reduced the oscillations induced by high K$^+$ PSS depolarization but had no effect on the steady-state (slow) response. Activation of TRP channels would be expected to increase the steady-state (slow) response. Inhibition of K$^+$ channels would be expected to have a minimal effect on high K$^+$ PSS-induced depolarization. Regarding inhibition of Orai by CCCP, we have previously examined the store content in smooth muscle cells before and after inhibition of Ca$^{2+}$ release by CCCP; no difference in the internal store Ca$^{2+}$ content occurred (McCarron & Muir, 1999).

In this study we have shown that there are two components to the signals resulting from voltage-dependent Ca$^{2+}$ entry. The first is a slow, persistent increase in the cytoplasmic concentration of the ion as expected from influx of Ca$^{2+}$ and the requirement for a uniform contraction to occur. The second component of the Ca$^{2+}$ signal is repetitive Ca$^{2+}$ oscillations and waves that occur as a result of Ca$^{2+}$ release from the internal store via IP$_3$R. We also show mitochondria exert little influence on the slow, persistent Ca$^{2+}$ increase that occurs as a direct result of voltage-dependent Ca$^{2+}$ entry. This result might be expected given that mitochondria are positioned at a distance from the plasma membrane in mesenteric artery smooth muscle cells (Firth et al., 2009). On the contrary mitochondria exert a profound influence on Ca$^{2+}$ release via IP$_3$R. Although our results are limited to influx via voltage-dependent Ca$^{2+}$ channels, it seems possible that Ca$^{2+}$ influx via any channel may trigger the process (Heathcote et al., 2019). The findings reveal a new complexity to the regulation of voltage-dependent Ca$^{2+}$ signals in vascular smooth muscle.

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

## Additional information

### Data availability statement

All study data are included in the article and/or supporting information.

## Competing interests

The authors declare that they have no competing interests.

## Author contributions

The final version of the manuscript has been approved by all authors (X.Z., C.B., M.D.L., S.C., C.W., J.G.M.), who also accept responsibility for all aspects of the work. Every individual listed as an author meets the criteria for authorship, and all who meet these criteria are included.

## Funding

This work was funded by the British Heart Foundation (RG/F/20/110007; PG/20/9/34859), whose support is gratefully acknowledged.

## Acknowledgements

The authors would like to thank Margaret MacDonald for her excellent technical support. The authors gratefully acknowledge the Beatson Advanced Imaging Resource at the CRUK Scotland Institute, in particular Dr Nikki Paul and Peter Thomason, for their support and assistance in this work.

## Keywords

inositol triphosphate receptors, mitochondria, smooth muscle cells, voltage-dependent $Ca^{2+}$ channels

## Supporting information

Additional supporting information can be found online in the Supporting Information section at the end of the HTML view of the article. Supporting information files available:

**Peer Review History**
**Supplementary Figures**
**Supplementary Video**

