## [Peer Review History · The Journal of Physiology]

Mitochondria regulate IP₃-mediated Ca²⁺ release triggered by voltage-dependent Ca²⁺ entry in resistance arteries

Xun Zhang, Charlotte Buckley, Matthew D Lee, Susan Chalmers, Calum Wilson, and John G McCarron

DOI: 10.1113/JP288022

Corresponding author(s): John McCarron (john.mccarron@strath.ac.uk)

The following individual(s) involved in review of this submission have agreed to reveal their identity: Ken D O'Halloran (Referee #3); Sean Williams (Referee #4)

Review Timeline:

Submission Date:	31-Oct-2024
Editorial Decision:	28-Nov-2024
Revision Received:	19-Dec-2024
Editorial Decision:	27-Jan-2025
Revision Received:	27-Feb-2025
Editorial Decision:	10-Mar-2025
Revision Received:	11-Mar-2025
Accepted:	19-Mar-2025

Senior Editor: Bjorn Knollmann

Reviewing Editor: Nikki Jernigan

Transaction Report:

Dear Dr McCarron,

Re: JP-RP-2024-288022 "Mitochondria regulate IP3-mediated Ca²⁺ release triggered by voltage-dependent Ca²⁺ entry in resistance arteries" by Xun Zhang, Charlotte Buckley, Matthew D Lee, Susan Chalmers, Calum Wilson, and John G McCarron

Thank you for submitting your manuscript to The Journal of Physiology. It has been assessed by a Reviewing Editor and by 3 expert referees and we are pleased to tell you that it is potentially acceptable for publication following satisfactory major revision.

LANGUAGE EDITING AND SUPPORT FOR PUBLICATION: If you would like help with English language editing, or other article preparation support, Wiley Editing Services offers expert help, including English Language Editing, as well as translation, manuscript formatting, and figure formatting at www.wileyauthors.com/eoo/preparation. You can also find resources for Preparing Your Article for general guidance about writing and preparing your manuscript at www.wileyauthors.com/eoo/prepresources.

REVISION CHECKLIST:

We look forward to receiving your revised submission.

Yours sincerely,

Bjorn Knollmann
Senior Editor
The Journal of Physiology

REQUIRED ITEMS

- Author photo and profile. First or joint first authors are asked to provide a short biography (no more than 100 words for one author or 150 words in total for joint first authors) and a portrait photograph. These should be uploaded and clearly labelled together in a Word document with the revised version of the manuscript. See Information for Authors for further details.

- Your manuscript must include a complete Additional Information section, including competing interests; funding; author contributions and acknowledgements.

- Please upload separate high-quality figure files via the submission form.

- Papers must comply with the Statistics Policy: https://jp.msubmit.net/cgi-bin/main.plex?form_type=display_requirements#statistics.

In summary:

- If $n \leq 30$, all data points must be plotted in the figure in a way that reveals their range and distribution. A bar graph with data points overlaid, a box and whisker plot or a violin plot (preferably with data points included) are acceptable formats.

- If $n > 30$, then the entire raw dataset must be made available either as supporting information, or hosted on a not-for-profit repository, e.g. FigShare, with access details provided in the manuscript.

- 'n' clearly defined (e.g. x cells from y slices in z animals) in the Methods. Authors should be mindful of pseudoreplication.

- All relevant 'n' values must be clearly stated in the main text, figures and tables.

- The most appropriate summary statistic (e.g. mean or median and standard deviation) must be used. Standard Error of the Mean (SEM) alone is not permitted.

- Exact p values must be stated. Authors must not use 'greater than' or 'less than'. Exact p values must be stated to three significant figures even when 'no statistical significance' is claimed.

- Please include an Abstract Figure file, as well as the Figure Legend text within the main article file. The Abstract Figure is a piece of artwork designed to give readers an immediate understanding of the research and should summarise the main conclusions. If possible, the image should be easily 'readable' from left to right or top to bottom. It should show the

physiological relevance of the manuscript so readers can assess the importance and content of its findings. Abstract Figures should not merely recapitulate other figures in the manuscript. Please try to keep the diagram as simple as possible and without superfluous information that may distract from the main conclusion(s). Abstract Figures must be provided by authors no later than the revised manuscript stage and should be uploaded as a separate file during online submission labelled as File Type 'Abstract Figure'. Please also ensure that you include the figure legend in the main article file. All Abstract Figures should be created using BioRender. Authors should use The Journal's premium BioRender account to export high-resolution images. Details on how to use and access the premium account are included as part of this email.

EDITOR COMMENTS

Reviewing Editor:

Ethics Concerns:

You must respond to the Comments of the Ethics reviewer (Referee 3)

Comments for Authors to ensure the paper complies with the Statistics Policy (Required):

Need to provide precise p values.

Comments to the Author :

The study offers a novel perspective on intracellular calcium regulation in vascular smooth muscle cells, focusing on interactions between L-type voltage-gated calcium channels (VGCCs), IP3 receptors, and mitochondria. However, there are two major concerns. One is lack of evidence that L-VGCCs mediate the sustained calcium influx compared to other mechanisms. Additional experiments can be completed to easily address this. The other issue is the reliance on drugs with non-specific effects, which are not adequately addressed.

Please also see 'Required Items' above.

Senior Editor:

Comments for Authors to ensure the paper complies with the Statistics Policy (Required):

Please comply with the journals statistics policy

Comments to the Author:

The reviewers and reviewing editor found potential merit in the work, but two major concerns were raised that would have to be addressed experimentally, as pointed out by the reviewers.

REFEREE COMMENTS

Referee #1:

1. The regulation of cytoplasmic calcium is critical for many cells, and studies attempting to shed light on this issue for smooth muscle cells go back many decades. The present study progresses our knowledge in the area by investigating the interactions between the endo/sarcoplasmic reticulum and the mitochondria. Importantly, the authors address this issue by studying intact tissues, in this case small resistance arteries from the rat. The main technique involves the use of imaging of cytoplasmic free calcium and also of changes in the mitochondrial membrane potential. The results appear clear cut and support the conclusions. However, the study is weakened by its strong reliance on drugs that are known to have a variety of non-specific effects.

2. The English expression needs some attention since there are words missing in various places and the wrong tense in places. It is noted that American spelling and words are used, not those of the UK.

3. Lines 179-180: The tissues were being superfused, not perfused. The heading and subsequent text need to be changed.

4. Lines 180-188: The temperature of the superfusing solutions need to be included in this section.
5. Line 220: "Sylgard coated" may be better than "Sylgard mounted".
6. Lines 287-288: "data was" should be "data were" since data is plural.
7. Line 295: "high-speed microscopy". This is somewhat misleading since the data were acquired at 10 Hz, whereas many readers may associate "high-speed" with much higher speeds of acquisition.
8. Lines 380-381: ".. the inhibition was rescued ...". This is not overly clear. Better to re-phrase along the lines of " .. the calcium signal was rescued ..".
9. In various places, particularly in the figure legends, the unit of distance is abbreviated as "M". In the SI system, metre is abbreviated with the lower case version, that is "m". This is particularly confusing since various of the figures deal with both concentrations (hence "M" for Molar), and distance (for which "M" is not the unit).
10. The authors rely on some drugs that are notorious for their non-specific effects. These include 2-APB, U73122 and CCCP. Drumm et al, (Cell Calcium 2018) state that " Protonophores can have off target effects, including indirect effects on the Na/K pump [37], depolarization due to the activation of proton Na⁺ pumps [38], impairment of lysosomal function [39] and changes in intracellular pH and generation of free radicals [40]." The reported effects of 2-APB include the activation of some TRP channels and inhibition of Orai and some K⁺ channels, inhibition of NOX, and inhibition of calcium uptake by mitochondria. These effects are very relevant to this study.
11. The Discussion should include acknowledgement that some of the drugs used have non-specific effects, even if not relevant to this study since such acknowledgement gives the reader greater confidence that the authors are aware of these issues and may have taken that into account, thereby giving greater confidence in the results. Furthermore, readers may use these drugs in their own studies and assume that they are OK.
12. Following on from point 10, in Fig 4C, for the two left sets of data (spontaneous oscillations) the effect of U73122 is "ns". This raises two issues. First, since there is an appreciable difference in the mean values, it could be that U73122 was inhibiting K⁺ channels, one of its reported non-specific effects. This would tend to depolarize the cells, thereby causing an increase in activity and explaining this difference in means. Second, it seems likely that the p value must be only just larger than 0.05. It would be more meaningful to the reader if the actual value of p is given to four decimal places so that they can make up their own minds as to whether they consider the difference in means to be statistically significant or not. This should apply to all p-values.
13. In Fig 5A, the red data line showing TMRE fluorescence for the mitochondrial membrane potential, is relatively flat. However, there is no indication as to scale so it is not clear if there were no depolarizations associated with rises in cytoplasmic calcium. There are some small bumps in the red trace that are associated with peaks in the green trace, but the lack of a scale makes it difficult to know whether they are significant or not.
14. Fig 6D: The y-scale is "Number of fast Ca²⁺ events". It would be interesting to know over what time period.
15. Line 768, Fig 4 legend: "For summary data (C, D, E), ...". Should be (B, C) since there are no panels D, E.

16. In Fig 2, the slow response is much larger than the fast response, whereas in Fig 3, the opposite is the case. What may be the reason(s) for these differences in responses?

17. Lines 493 - 502. In this paragraph within the Discussion, the authors consider the lack of mitochondrial depolarization during calcium uptake and comment that other studies have found a similar result. The authors should also include in their discussion reference to the paper by Yamamura et al (Am J Physiol 2018, 314:C88-C98) who show very elegantly that mitochondria in smooth muscle depolarize during calcium uptake. Furthermore, Yamamura et al show that block of mitochondrial calcium uptake is associated with an increase in cytoplasmic calcium, unlike the lack of effect in the authors' Fig 6E.

Referee #2:

The study is aimed to investigate the interactions between Ca influx via L-type voltage-gated Ca channels (VGCCs), IP3-dependent Ca-release and its regulation by mitochondria. The group has an established national and international reputation and expertise looking at individual aspects of intracellular calcium signalling which involves calcium release from intracellular stores via IP3Rs and RyRs and the regulatory role of mitochondria in these processes. The topic complex due to existence of intracellular compartmentalisation and microdomains that can vary in different vascular beds. The MS is well written and carefully presented.

General comments:

1. One of my main criticisms is the main conclusion (Discussion 1st paragraph and Fig. 9) that implies a unique role of VGCCs as the sources of a sustained elevated Ca as a trigger for IP3-release its modulation by mitochondria. I feel this is quite ambiguous statement with no direct evidence to support the key role for VGCCs or the requirement of VGCC to be active to trigger it. One can argue that any submaximal sustained raise of cytosolic calcium can trigger it. This is fundamental question since under physiological conditions it is more likely that both agonist-mediated IP3 release and VGCC Ca influx are contributing to basal tone. The results with caged IP3 (Fig. 8) are not really supportive as they lack a sustained component which is apparently required for the described effects. Likewise, PLC inhibitor wouldn't be expected to have any effect (Fig. 4) as there is no agonist to activate its activity. I personally feel that it is physiologically important to demonstrate that an agonist (e.g. phenylephrine) at a low concentration that cause a comparable basal increase in calcium but in the presence of VGCC inhibitor, will not show oligomycin-sensitive fast calcium oscillations. Even if it does, this will not diminish the impact of findings just tune the conclusion.

2. Discussion of Fig. 5 regarding to the proximity of mitochondria to the cell membrane is ambiguous. The study from Firth et al, (2009, doi: 10.1152/ajplung.90341.2008) showed that in the rat mesenteric smooth muscle cells mitochondria are distinctively away from the cell membrane, and more closely associated with the SR deeper inside the cell which is supportive of your concept of their interactions with IP3 receptors. Such juxtaposition, however, could be different in other vascular beds, e.g. in pulmonary vasculature. These aspects need to be properly discussed.

3. Methods. Line 151. This is not adequate description. Please provide dose of anaesthetic used and the Schedule 1 method used.

Minor comments:

Line 104 & 343. Grammatical error in few places: "mitochondria's" use either "...of mitochondria" or "mitochondrial..." as appropriate.

Line 299 typo: " to be to two"

Referee #3 (ethics review):

Thank you for submitting your manuscript to The Journal of Physiology.

Some additional details pertaining to animal welfare are required.

1. You must begin the Methods section with the subheading "Ethical approval". If a specific approval code was provided for the study then please provide this in the text.
2. Line 151: should be "euthanised".
3. Please confirm the body weight of the rats. At the reported age range 10-12 weeks, they are likely to have weighed >150g. Rodents >150g must be sedated or anaesthetised prior to cervical dislocation. Please elaborate on the full fate of the animals.

END OF COMMENTS

Bjorn Knollmann
Senior Editor
The Journal of Physiology

Dear Dr Knollmann

Thank you, the Editors and the Referees for your time and consideration of our manuscript. We have carefully considered each of the reviewers' comments made significant revisions to address their concerns. In our rebuttal, we have provided detailed responses to each point raised by the reviewers and explain the changes made in the revised manuscript to address these issues. We believe that the revisions have substantially improved the manuscript.

Once again, we would like to express our gratitude for the opportunity to submit our work to Journal of Physiology and for the helpful feedback provided by the Editor and Reviewers.

Yours Sincerely,

John G McCarron
W.C. Bowman Chair of Pharmacology

Reviewing Editor:

We thank the Reviewing Editor for the careful and thoughtful consideration of our manuscript. We have addressed each point fully below.

COMMENT Ethics Concerns:

You must respond to the Comments of the Ethics reviewer (Referee 3)

RESPONSE Done

**COMMENT Comments for Authors to ensure the paper complies with the Statistics Policy (Required):
Need to provide precise p values.**

RESPONSE Done. Precise p values are now included on each figure.

COMMENT Comments to the Author :

The study offers a novel perspective on intracellular calcium regulation in vascular smooth muscle cells, focusing on interactions between L-type voltage-gated calcium channels (VGCCs), IP3 receptors, and mitochondria. However, there are two major concerns. One is lack of evidence that L-VGCCs mediate the sustained calcium influx compared to other mechanisms. Additional experiments can be completed to easily address this. The other issue is the reliance on drugs with non-specific effects, which are not adequately addressed.

RESPONSE The sustained Ca²⁺ increase is triggered by high K⁺-induced depolarization, and this increase is blocked by the voltage-dependent Ca²⁺ channel blocker nimodipine or by removing external Ca²⁺ (Figure 2). These findings strongly support our conclusion that the depolarisation evoked by high K⁺ PSS induces Ca²⁺ influx through voltage-dependent Ca²⁺ channels.

Furthermore, the sustained Ca²⁺ rise is unaffected by 2-APB, dantrolene, or store depletion with ryanodine and caffeine (Figure 3), providing clear evidence that Ca²⁺ release from intracellular stores does not contribute to the response. Additionally, high K⁺ PSS reduces the driving force for Ca²⁺ entry across the plasma membrane, which would decrease (rather than increase) Ca²⁺ influx through non-voltage-gated channels.

We recognise that the original manuscript may not have made it sufficiently clear that these findings (taken together) establish that the sustained Ca²⁺ rise triggered by high K⁺ depolarization occurs via voltage-dependent Ca²⁺ channels. To address this, we have revised the discussion to emphasise why the existing data fully support the conclusion.

Regarding the reliance on drugs with non-specific effects, which precisely were not adequately addressed, we have now added a substantial discussion of the drugs off target effects on page 21 of the manuscript.

.....

Senior Editor:

We thank the Senior Editor for the careful and thoughtful consideration of our manuscript. We have addressed each point fully below

COMMENT Comments for Authors to ensure the paper complies with the Statistics Policy (Required):
Please comply with the journals statistics policy

RESPONSE Done. All graphs show original data points, exact p values are reported in the figures, all n numbers (biological replicates) are reported, and statistical tests used are defined.

COMMENT Comments to the Author:

The reviewers and reviewing editor found potential merit in the work, but two major concerns were raised that would have to be addressed experimentally, as pointed out by the reviewers.

RESPONSE We thank the Editor and Referees' for the constructive and helpful comments. We have addressed each comment in the text below.

Referee #1:

We thank the Referee for the careful and thoughtful consideration of our manuscript. We have addressed each point fully below

COMMENT 1. The regulation of cytoplasmic calcium is critical for many cells, and studies attempting to shed light on this issue for smooth muscle cells go back many decades. The present study progresses our knowledge in the area by investigating the interactions between the endo/sarcoplasmic reticulum and the mitochondria. Importantly, the authors address this issue by studying intact tissues, in this case small resistance arteries from the rat. The main technique involves the use of imaging of cytoplasmic free calcium and also of changes in the mitochondrial membrane potential. The results appear clear cut and support the conclusions. However, the study is weakened by its strong reliance on drugs that are known to have a variety of non-specific effects.

RESPONSE We thank the Referees for the supportive and helpful comments. We have addressed each comment in the text below.

COMMENT 2. The English expression needs some attention since there are words missing in various places and the wrong tense in places. It is noted that American spelling and words are used, not those of the UK.

RESPONSE Corrected

COMMENT 3. Lines 179-180: The tissues were being superfused, not perfused. The heading and subsequent text need to be changed.

RESPONSE Corrected

COMMENT 4. Lines 180-188: The temperature of the superfusing solutions need to be included in this section.

RESPONSE Done (room temperature 20°C; page 8).

COMMENT 5. Line 220: "Sylgard coated" may be better than "Sylgard mounted".

RESPONSE Corrected

COMMENT 6. Lines 287-288: "data was" should be "data were" since data is plural.

RESPONSE Corrected

COMMENT 7. Line 295: "high-speed microscopy". This is somewhat misleading since the data were acquired at 10 Hz, whereas many readers may associate "high-speed" with much higher speeds of acquisition.

RESPONSE Removed. Now reads "To investigate voltage-dependent Ca^{2+} entry in intact vascular smooth muscle cells, intracellular Ca^{2+} signals were examined in second order rat mesenteric arteries (Figure 1A)."

COMMENT 8. Lines 380-381: ".. the inhibition was rescued ...". This is not overly clear. Better to re-phrase along the lines of " .. the calcium signal was rescued ..".

RESPONSE Done

COMMENT 9. In various places, particularly in the figure legends, the unit of distance is abbreviated as "M". In the SI system, metre is abbreviated with the lower case version, that is "m". This is particularly confusing since various of the figures deal with both concentrations (hence "M" for Molar), and distance (for which "M" is not the unit).

RESPONSE Corrected

COMMENT 10. The authors rely on some drugs that are notorious for their non-specific effects. These include 2-APB, U73122 and CCCP. Drumm et al, (Cell Calcium 2018) state that " Protonophores can have off target effects, including indirect effects on the Na/K pump [37], depolarization due to the activation of proton Na⁺ pumps [38], impairment of lysosomal function [39] and changes in intracellular pH and generation of free radicals [40]." The reported effects of 2-APB include the activation of some TRP channels and inhibition of Orai and some K⁺ channels, inhibition of NOX, and inhibition of calcium uptake by mitochondria. These effects are very relevant to this study.

RESPONSE See response to Comment 11

COMMENT 11. The Discussion should include acknowledgement that some of the drugs used have non-specific effects, even if not relevant to this study since such acknowledgement gives the reader greater confidence that the authors are aware of these issues and may have taken that into account, thereby giving greater confidence in the results. Furthermore, readers may use these drugs in their own studies and assume that they are OK.

RESPONSE We have now added a substantial section Discussion (top of page 21) which points out the drugs off-target effects. The section reads:-

“Studies examining the role of mitochondria in regulating Ca²⁺ signalling often use pharmacological agents like protonophores to alter mitochondrial function. However, as Drumm et al. (<https://doi.org/10.1016/j.ceca.2018.01.003>) highlight, protonophores can have effects that extend beyond the mitochondria, at least under certain conditions. These off-target effects include the indirect inhibition of the sodium pump due to ATP depletion (<https://doi.org/10.1124/mol.53.4.734>), plasma membrane depolarization through activation of H⁺ and Na⁺ currents (<https://doi.org/10.1007/s004240100703>), changes in intracellular pH, and the generation of free radicals (<https://doi.org/10.1111/micc.12039>). Each of these effects will lead to steady-state changes in cell function.

In our study, the protonophore CCCP inhibited repetitive Ca²⁺ oscillations but did not alter the steady-state (slow) response. This suggests that off-target effects are unlikely to contribute to our findings. In support, some of the reported off-target effects, such as those on the sodium pump, stem from the specific action of protonophores on ATP synthesis inhibition by collapse of the mitochondrial membrane potential. In our experiments, inhibition of ATP synthesis by oligomycin (in the presence of antioxidants) had no effect on oscillations, and the effect of CCCP was unchanged by antioxidants. This further supports the conclusion that off-target effects of protonophores are not likely to be responsible for our findings.

The widely used IP₃ receptor blocker 2-APB is also reported to have off-target effects. These include activating some TRP channels or inhibiting Orai, K⁺ channels, and mitochondrial Ca²⁺ uptake. None of these off-target effects explain the present findings. In the present study, 2-APB reduced the oscillations induced by high K⁺ PSS depolarization but had no effect on the steady-state (slow) response. Activation of TRP channels would be expected to increase the steady-state (slow) response. Inhibition of K⁺ channels would be expected to have minimal effect on high K⁺ PSS induced depolarization.

Regarding inhibition of Orai by CCCP. We have previously examined the store content in smooth muscle cells before and after inhibition of Ca²⁺ release by CCCP and seen no difference in the internal store Ca²⁺

content (<https://doi.org/10.1111/j.1469-7793.1999.149aa.x>).”

COMMENT 12. Following on from point 10, in Fig 4C, for the two left sets of data (spontaneous oscillations) the effect of U73122 is "ns". This raises two issues. First, since there is an appreciable difference in the mean values, it could be that U73122 was inhibiting K⁺ channels, one of its reported non-specific effects. This would tend to depolarize the cells, thereby causing an increase in activity and explaining this difference in means. Second, it seems likely that the p value must be only just larger than 0.05. It would be more meaningful to the reader if the actual value of p is given to four decimal places so that they can make up their own minds as to whether they consider the difference in means to be statistically significant or not. This should apply to all p-values.

RESPONSE The referee is referring the effects of U73122 on basal Ca²⁺ spiking activity. We have now shown the p value on the figure and it is far from significance (p=0.2562).

COMMENT 13. In Fig 5A, the red data line showing TMRE fluorescence for the mitochondrial membrane potential, is relatively flat. However, there is no indication as to scale so it is not clear if there were no depolarizations associated with rises in cytoplasmic calcium. There are some small bumps in the red trace that are associated with peaks in the green trace, but the lack of a scale makes it difficult to know whether they are significant or not.

RESPONSE The scale bar is given and an expanded inset is now shown. The small variations in fluorescence are likely to have occurred because of small movements of the smooth muscle cells. In the present experiments, depolarization of mitochondria would generate a decrease in TMRE fluorescence as we are using the indicator below its quench limit (therefore depolarization is associated with a reduction in concentration of TMRE in mitochondria).

We have previously studied the relationship between mitochondrial membrane potential and Ca²⁺ signals generated by Ca²⁺ release via (1) IP₃ receptors or (2) ryanodine receptors or (3) Ca²⁺ influx via voltage-dependent Ca²⁺ channels in smooth muscle cells; none generated a detectable change in mitochondrial membrane potential (e.g. <https://doi.org/10.1242/jcs.014522>). In those studies, the limit of resolution was ~5 mV in mitochondrial membrane potential.

Importantly, if increases in Ca²⁺ significantly depolarized the mitochondrial membrane potential there would be significant detrimental consequences to cell function. Minimally, a cell's ability to generate ATP would be compromised and there would be a substantial change in ROS production. Each of these changes may lead to unexpected changes in cell function as (for example) an increase in Ca²⁺ triggers contraction which requires ATP. Depolarization of the mitochondrial membrane potential is also associated with apoptosis.

As we pointed out in the Discussion section “Two explanations may account for the absence of a measured mitochondrial depolarisation. The current carried by Ca²⁺ may be insufficient to alter the inner mitochondrial membrane potential, or there may be a compensatory efflux of positively charged species (such as protons) counteracting the depolarization induced by Ca²⁺ influx. Similar to our findings, other studies have also failed to detect significant mitochondrial membrane potential depolarization in response to transient increases in cytoplasmic Ca²⁺ concentrations induced by IP₃-generating agonists, observed in cell types like HeLa or RBL-2H3 cells (Collins et al., 2000; Hajnóczky et al., 2003).”

COMMENT 14. Fig 6D: The y-scale is "Number of fast Ca²⁺ events". It would be interesting to know over what time period.

RESPONSE It is the averaged number of events over the entire analysis period of ten minutes. This is now clarified in the figure caption.

COMMENT 15. Line 768, Fig 4 legend: "For summary data (C, D, E), ...". Should be (B, C) since there are no panels D, E.

RESPONSE Corrected

COMMENT 16. In Fig 2, the slow response is much larger than the fast response, whereas in Fig 3, the opposite is the case. What may be the reason(s) for these differences in responses?

RESPONSE Thank you for pointing this out. The fast and slow responses were mislabelled we have now corrected the labelling.

COMMENT 17. Lines 493 - 502. In this paragraph within the Discussion, the authors consider the lack of mitochondrial depolarization during calcium uptake and comment that other studies have found a similar result. The authors should also include in their discussion reference to the paper by Yamamura et al (Am J Physiol 2018, 314:C88-C98) who show very elegantly that mitochondria in smooth muscle depolarize during calcium uptake. Furthermore, Yamamura et al show that block of mitochondrial calcium uptake is associated with an increase in cytoplasmic calcium, unlike the lack of effect in the authors' Fig 6E.

RESPONSE The reference is now included (page 20 line 10).

Changes in mitochondrial membrane potential during Ca^{2+} signals have been reported in some studies but not in others. Our experience in several studies (e.g. <https://doi.org/10.1242/jcs.014522>) is that cytoplasmic Ca^{2+} increases arises from depolarization evoked Ca^{2+} entry, IP_3 evoked Ca^{2+} release or release of Ca^{2+} from the store via RyR does not alter the mitochondrial membrane potential. Depolarization of the mitochondria membrane potential would compromise ATP production that at a time when ATP was required to support contraction.

The relationship between block of mitochondrial Ca^{2+} uptake and the peak Ca^{2+} response has been also investigated in several studies and is variously reported as increased or decreased or unchanged depending on the nature of the Ca^{2+} signal. Our experience in several studies (e.g. <https://doi.org/10.1242/jcs.014522>; <https://doi.org/10.1111/j.1469-7793.1999.149aa.x>; [https://www.jbc.org/article/S0021-9258\(19\)64911-2/fulltext](https://www.jbc.org/article/S0021-9258(19)64911-2/fulltext); <https://doi.org/10.1021/ja2077922>; <https://doi.org/10.1016/j.ceca.2009.05.007>) is that the Ca^{2+} rise from IP_3 -evoked Ca^{2+} release is decreased but there is no significant change in the peak Ca^{2+} change arising from voltage-dependent Ca^{2+} entry (which appears also to be what Yamamura et al found) although Ca^{2+} removal from the cytoplasmic was slowed (which appears also to be what Yamamura et al found).

We thank the Referee again for the very helpful and constructive comments and hope we have addressed each point fully.

Referee #2:

We thank the Referee for the careful and thoughtful consideration of our manuscript. We have addressed each point fully below

COMMENT The study is aimed to investigate the interactions between Ca influx via L-type voltage-gated Ca channels (VGCCs), IP₃-dependent Ca-release and its regulation by mitochondria. The group has an established national and international reputation and expertise looking at individual aspects of intracellular calcium signalling which involves calcium release from intracellular stores via IP₃Rs and RyRs and the regulatory role of mitochondria in these processes. The topic complex due to existence of intracellular compartmentalisation and microdomains that can vary in different vascular beds. The MS is well written and carefully presented.

RESPONSE We thank the Referee for the supportive and helpful comments.

General comments:

COMMENT 1. One of my main criticisms is the main conclusion (Discussion 1st paragraph and Fig. 9) that implies a unique role of VGCCs as the sources of a sustained elevated Ca as a trigger for IP₃-release its modulation by mitochondria. I feel this is quite ambiguous statement with no direct evidence to support the key role for VGCCs or the requirement of VGCC to be active to trigger it. One can argue that any submaximal sustained raise of cytosolic calcium can trigger it. This is fundamental question since under physiological conditions it is more likely that both agonist-mediated IP₃ release and VGCC Ca influx are contributing to basal tone. The results with caged IP₃ (Fig. 8) are not really supportive as they lack a sustained component which is apparently required for the described effects. Likewise, PLC inhibitor wouldn't be expected to have any effect (Fig. 4) as there is no agonist to activate its activity. I personally feel that it is physiologically important to demonstrate that an agonist (e.g. phenylephrine) at a low concentration that cause a comparable basal increase in calcium but in the presence of VGCC inhibitor, will not show oligomycin-sensitive fast calcium oscillations. Even if it does, this will not diminish the impact of findings just tune the conclusion.

RESPONSE The response we were studying was triggered by a high K⁺ PSS bath solution and blocked by the voltage-dependent Ca²⁺ channel blocker nimodipine or by removing external Ca²⁺ (Figure 2). These findings suggest that depolarisation evoked by high K⁺ PSS induces Ca²⁺ influx via voltage-dependent Ca²⁺ channels. The sustained Ca²⁺ rise evoked by high K⁺ PSS was unaffected by 2-APB, dantrolene, or store depletion with ryanodine and caffeine (Figure 3), indicating that Ca²⁺ release from intracellular stores does not contribute to the response. Additionally, the depolarisation evoked by high K⁺ PSS will reduce the driving force for Ca²⁺ entry across the plasma membrane, which would decrease (rather than increase) Ca²⁺ influx through non-voltage-gated channels. The simplest explanation of the results is that the sustained Ca²⁺ rise triggered by high K⁺ depolarization occurs via voltage-dependent Ca²⁺ channels.

By no means do we intend to claim that influx via voltage-dependent Ca²⁺ channels is the only way the IP₃ receptor may be activated. We completely agree that any Ca²⁺ influx is likely to trigger the process. Indeed, we have previously reported that Ca²⁺ influx via TRPV4, activated Ca²⁺ release via IP₃ receptors in endothelial cells (<https://doi.org/10.1111/bph.14762>). However, for the reasons outlined above, this is unlikely in the present experimental conditions. We have now clarified this point in the Discussion section (second last section page 22) which reads "While our results are limited to influx via voltage-dependent Ca²⁺ channels, it seems possible that Ca²⁺ influx via any channel may trigger the process (<https://doi.org/10.1111/bph.14762>)."

Phenylephrine activates Ca²⁺ release from the internal store via IP₃ receptors. We have previously found that the response to phenylephrine was largely blocked and reduced to a transient contraction when the internal store was depleted of Ca²⁺. This makes the Referee's suggested experiment to examine the effect of phenylephrine as a trigger for Ca²⁺ influx difficult to achieve.

COMMENT 2. Discussion of Fig. 5 regarding to the proximity of mitochondria to the cell membrane is ambiguous. The study from Firth et al, (2009, doi: 10.1152/ajplung.90341.2008) showed that in the rat mesenteric smooth muscle cells mitochondria are distinctively away from the cell membrane, and more closely associated with the SR deeper inside the cell which is supportive of your concept of their interactions with IP3 receptors. Such juxtaposition, however, could be different in other vascular beds, e.g. in pulmonary vasculature. These aspects need to be properly discussed.

RESPONSE We have removed the suggestion that the organelles are largely positioned near the plasma membrane as it is not supported fully in the data presented. Our previous studies in which we used the electrical signature of the mitochondria (recorded as 'flickers' of from fluctuations in the fluorophore TMRE intensity) to measure the organelles shape and position did find the organelles were largely positioned near the plasma membrane (FaLM; <https://doi.org/10.1038/srep16875>)

COMMENT 3. Methods. Line 151. This is not adequate description. Please provide dose of anaesthetic used and the Schedule 1 method used.

RESPONSE Animals were euthanised by cervical dislocation. **Please note:-**

Sedation or anesthesia prior to cervical dislocation may be used but is not a requirement. The UK ASPA has the following exception:-

"Nothing in this Schedule requires or permits the prior use of sedative or anaesthetic where the distress likely to be caused by administering it is greater than the distress likely to be caused by using the appropriate method of killing without sedative or anaesthetic."

At Strathclyde the NVS and NACWO deem that the use of sedative increases the stress to the animal and provided an experience technician undertakes the Schedule 1 we do not sedate them prior to rapid cervical dislocation, followed by a confirmation method.

Minor comments:

COMMENT Line 104 & 343. Grammatical error in few places: "mitochondria's" use either "...of mitochondria" or "mitochondrial..." as appropriate.

RESPONSE Corrected

COMMENT Line 299 typo: " to be to two"

RESPONSE Corrected

We thank the Referee again for the very helpful and constructive comments and hope we have addressed each point fully.

Referee #3 (ethics review):

We thank the Referee for the careful and thoughtful consideration of our manuscript. We have addressed each point fully below

Thank you for submitting your manuscript to The Journal of Physiology. Some additional details pertaining to animal welfare are required.

COMMENT 1. You must begin the Methods section with the subheading "Ethical approval". If a specific approval code was provided for the study then please provide this in the text.

RESPONSE Done. Methods now begins with:-

"Ethical approval

The University of Strathclyde holds Establishment license number X56B4FB08 under The Animal (Scientific Procedures) Act 1986. It is an underlying principle of the Animal (Scientific Procedures) Act 1986 that animals bred, supplied and used for scientific procedures are cared for in accordance with the best standards of modern animal husbandry and comply with the Code of Practice on Care and Accommodation."

COMMENT 2. Line 151: should be "euthanised".

RESPONSE Corrected

COMMENT 3. Please confirm the body weight of the rats. At the reported age range 10-12 weeks, they are likely to have weighed >150g. Rodents >150g must be sedated or anaesthetised prior to cervical dislocation. Please elaborate on the full fate of the animals.

RESPONSE The body weight is in the region of 300-350g.

Sedation or anesthesia prior to cervical dislocation may be used but is not a requirement. The UK ASPA has the following exception:-

"Nothing in this Schedule requires or permits the prior use of sedative or anaesthetic where the distress likely to be caused by administering it is greater than the distress likely to be caused by using the appropriate method of killing without sedative or anaesthetic."

At Strathclyde the NVS and NACWO deem that the use of sedative increases the stress to the animal and provided an experience technician undertakes the Schedule 1 we do not sedate them prior to rapid cervical dislocation, followed by a confirmation method.

We thank the Referee again for the very helpful and constructive comments and hope we have addressed each point fully.

Dear Dr McCarron,

Re: JP-RP-2024-288022R1 "Mitochondria regulate IP3-mediated Ca²⁺ release triggered by voltage-dependent Ca²⁺ entry in resistance arteries" by Xun Zhang, Charlotte Buckley, Matthew D Lee, Susan Chalmers, Calum Wilson, and John G McCarron

Thank you for submitting your manuscript to The Journal of Physiology. It has been assessed by a Reviewing Editor and by 3 expert referees and we are pleased to tell you that it is potentially acceptable for publication following satisfactory major revision.

REVISION CHECKLIST:

We look forward to receiving your revised submission.

Yours sincerely,

Bjorn Knollmann
Senior Editor
The Journal of Physiology

REQUIRED ITEMS

- Your manuscript must include a complete Additional Information section, including competing interests; funding; author contributions and acknowledgements.

- Papers must comply with the Statistics Policy: https://jp.msubmit.net/cgi-bin/main.plex?form_type=display_requirements#statistics.

In summary:

- If $n \leq 30$, all data points must be plotted in the figure in a way that reveals their range and distribution. A bar graph with data points overlaid, a box and whisker plot or a violin plot (preferably with data points included) are acceptable formats.

- If $n > 30$, then the entire raw dataset must be made available either as supporting information, or hosted on a not-for-profit repository, e.g. FigShare, with access details provided in the manuscript.

- 'n' clearly defined (e.g. x cells from y slices in z animals) in the Methods. Authors should be mindful of pseudoreplication.

- All relevant 'n' values must be clearly stated in the main text, figures and tables.

- The most appropriate summary statistic (e.g. mean or median and standard deviation) must be used. Standard Error of the Mean (SEM) alone is not permitted.

- Exact p values must be stated. Authors must not use 'greater than' or 'less than'. Exact p values must be stated to three significant figures even when 'no statistical significance' is claimed.

Reviewing Editor:

Comments to ensure the paper complies with the Statistics Policy:

Please review comments about statistical comparisons carefully. This may require consultation with a statistician and reevaluating the data which may change the interpretation of the data.

Comments to the authors:

While the authors have addressed several concerns, a few issues remain. The primary concern pertains to the statistical comparisons. Please carefully address the feedback from Reviewer #1 and the Statistics Editor, as this may necessitate reevaluating the data and revising the interpretation.

Senior Editor:

Comments to ensure the paper complies with the Statistics Policy:

Please address the statistical concerns

Comments to the authors:

I concur with the Reviewing Editor.

Referee #1:

1. The authors have now addressed the issues that were raised previously, thereby improving the manuscript. The inclusion of the actual values of p is most informative for the reader. However, there are several issues that require attention, of which the actual values of p are of concern.

2. With reference to the original point 12, in Fig 4 panel C, the left 2 sets of data now have a stated p-value, with the value of p being 0.2562. This value is quite close to that in panel B, right hand side sets of data, where the p-value is 0.2854, despite the much tighter spread of data in panel B and the much smaller difference in mean values. In view of this, the graph in panel C was blown up in size and the value of each data point was measured. There is of course an element of inaccuracy in such measurements. Nevertheless, an unpaired t-test gives a p-value of ~0.07, which is quite different to the value of 0.2562 in Fig 4C. Paired testing may result in a lower, statistically significant p-value and therefore an effect of U73122 and this would have implications in the interpretation of the data and its discussion. If the data are paired and tested as such, this should be stated in all relevant figure legends.

3. In a similar vein, in Fig 7C left hand graph, the stated p-value is 0.0499, which many readers may consider is too marginal to have any real meaning one way or the other. However, after measuring the individual data points, and given that the data are presented in a paired manner, a paired t-test gives a p-value of 0.006, which is very different to the stated value of 0.0499 and which means that the difference is very significant. Again, the type of statistical testing used should be stated in the figure legend.

4. Other p-values in the manuscript were not checked but it is suggested that the authors should check all p-values with great care and include details of the statistical testing in the figure legends.

5. With reference to the original point 2, there is still considerable room for improvement in the English expression.

Referee #2:

I am happy with the authors' responses to my main comment but have a couple minor comments on the two other points.

1. The authors stated in their response that in previous study using FaLM did find the organelles [mitochondria] were largely positioned near the plasma membrane (Chalmers et al, 2015, DOI: 10.1038/srep16875). However, that study compared the size, density and clustering of mitochondria in WKY and SHR rats, with no direct evidence that indicates their localisation with the cell membrane. Firth et al, 2009 quantitatively measured co-localisation by labelling the cell membrane, which, in my opinion, doesn't contradict but rather supports the authors' overall conclusion, between SR/mitochondria interactions shown in summary Figure 9.

2. Animal humane killing: I agree with what the authors said about the choice, but there is no mentioning of a conformational method, I presume exsanguination. This need to be mentioned in the methods.

Referee #4:

Comments for authors:

The statistical methods section describes the use of paired t-tests and one-way ANOVA for multi-group comparisons, with data reported as mean {plus minus} SD. While the overall approach is broadly appropriate, there are several key concerns that need to be addressed to ensure statistical rigor and the validity of the results:

The methods state that multiple comparisons were conducted following ANOVA but do not specify the correction method used. Without an appropriate post-hoc adjustment (e.g., Bonferroni), there is an increased risk of Type I errors due to multiple hypothesis testing. It is essential to clarify the post-hoc procedure employed and ensure that it appropriately controls for false positives. The authors also need to specify whether their ANOVA was a repeated-measures or independent one-way ANOVA.

The description indicates that data were extracted from a single preparation with different treatments. This raises the possibility of pseudoreplication if multiple measurements from the same biological sample were treated as independent replicates. It should be explicitly stated whether the replicates represent true biological replicates or technical replicates, as the latter would necessitate statistical adjustments (e.g., mixed-effects models or hierarchical analysis) rather than simple t-tests or ANOVA.

The use of a paired t-test assumes normality of differences between paired observations, which is particularly problematic given the small sample size ($n = 5$). With such a low n , normality tests have low power, and violations of normality can substantially impact results. A Wilcoxon signed-rank test would likely be a more robust alternative, providing similar power while avoiding issues related to non-normality. It is recommended that the authors explicitly report whether normality was tested and, if violated, confirm that a non-parametric alternative was used. As above, I would suggest going with non-parametric tests regardless of the outcomes of normality testing, given the very low N . Similarly, if repeated measures ANOVA was used, sphericity should be assessed via Mauchly's test, with Greenhouse-Geisser corrections applied where necessary.

While the statistical analysis section states that statistical significance was set at $p < 0.05$, it would be beneficial to supplement this with effect size measures (e.g., Cohen's d for t-tests, η^2 for ANOVA) to provide a better understanding of the magnitude of effects. Useful resource: <https://matthewbjane.quarto.pub/guide-to-effect-sizes-and-confidence-intervals/>. Additionally, GraphPad Prism's default statistical settings should be confirmed, as it may apply corrections automatically without explicit user input.

END OF COMMENTS

Dr Bjorn Knollmann
Senior Editor
The Journal of Physiology

Dear Dr Knollmann

Thank you, the Editors and the Referees for your time and consideration of our manuscript. We have carefully considered each of the Editor's and Reviewers' comments made significant revisions to address their concerns. In our rebuttal, we have carefully re-analyzed all statistical tests based on the recommendations. We checked the data for normality at each step and applied the appropriate statistical tests. Our manuscript now complies with the statistics policy. In brief, the final paragraph of the Methods section provides full details of the statistical treatment of the data, all data points are shown in the respective figures, mean and standard deviation of n independent biological replicates is used throughout, all n values are stated in the text and figure legends, and n is clearly defined throughout, all p values are reported to three significant figures.

In the rebuttals below, detailed responses to each point raised by the Editors and Reviewers are provided and we explain the changes made in the revised manuscript to address the points raised.

Once again, we would like to express our gratitude for the opportunity to submit our work to Journal of Physiology and for the helpful feedback provided by the Editor and Reviewers.

Yours Sincerely,

John G McCarron
W.C. Bowman Chair of Pharmacology

REQUIRED ITEMS

COMMENT - Your manuscript must include a complete Additional Information section, including competing interests; funding; author contributions and acknowledgements.

Response: The text on data availability, competing interests, author contributions, funding, and acknowledgements has been moved to the Additional Information Section.

COMMENT - Papers must comply with the Statistics Policy: https://jp.msubmit.net/cgi-bin/main.plex?form_type=display_requirements#statistics. In summary:

If n {less than or equal to} 30, all data points must be plotted in the figure in a way that reveals their range and distribution. A bar graph with data points overlaid, a box and whisker plot or a violin plot (preferably with data points included) are acceptable formats.

Response: All data point are shown in each figure.

COMMENT - If $n > 30$, then the entire raw dataset must be made available either as supporting information, or hosted on a not-for-profit repository, e.g. FigShare, with access details provided in the manuscript.

Response: Not applicable.

COMMENT - 'n' clearly defined (e.g. x cells from y slices in z animals) in the Methods. Authors should be mindful of pseudoreplication.

Response: The reported n value is the number of biological replicates (i.e., the number of animals). The following has been added to the Statistical Analysis section of the methods: *“A single artery from each animal contributed one data point per experiment. For all experiments, n represents the number of biological replicates (number of animals).”*

COMMENT - All relevant 'n' values must be clearly stated in the main text, figures and tables.

Response: All relevant n values, which represent the number of independent biological replicates (animals), are clearly states in the main text, as well as in the figure legends. Additionally, individual data points are shown in figures.

COMMENT - The most appropriate summary statistic (e.g. mean or median and standard deviation) must be used. Standard Error of the Mean (SEM) alone is not permitted.

Response: Mean and standard deviation of n independent biological replicates is used throughout.

COMMENT - Exact p values must be stated. Authors must not use 'greater than' or 'less than'. Exact p values must be stated to three significant figures even when 'no statistical significance' is claimed.

Response: All p values are reported in the respective figures to three significant figures. All data and test results are included in the supplementary data.

Reviewing Editor:

COMMENT - to ensure the paper complies with the Statistics Policy:

Please review comments about statistical comparisons carefully. This may require consultation with a statistician and reevaluating the data which may change the interpretation of the data.

Response: We have carefully reviewed all statistical test in the manuscript. Our manuscript now complies with the statistics policy as described above. In brief, the final paragraph of the methods section provides full details of the statistical treatment of the data, all data points are shown in the respective figures, mean and standard deviation of n independent biological replicates is used throughout, all n values are stated in the text and figure legends, and n is clearly defined throughout, all p values are reported to three significant figures.

Comments to the authors:

While the authors have addressed several concerns, a few issues remain. The primary concern pertains to the statistical comparisons. Please carefully address the feedback from Reviewer #1 and the Statistics Editor, as this may necessitate reevaluating the data and revising the interpretation.

Response: All p values are reported in the respective figures to three significant figures. All data and test results are included in the supplementary data.

Senior Editor:

Comments to ensure the paper complies with the Statistics Policy:

Please address the statistical concerns

Response: See response to the Reviewing Editor.

Comments to the authors:

I concur with the Reviewing Editor.

Response: See response to the Reviewing Editor.

Referee #1:

COMMENT 1. The authors have now addressed the issues that were raised previously, thereby improving the manuscript. The inclusion of the actual values of p is most informative for the reader. However, there are several issues that require attention, of which the actual values of p are of concern.

Response: Thank you for your careful attention and bringing this issue to our attention. In light of your comments, and those from the Editors, we have carefully reanalysed all data in the paper using appropriate statistical tests, and the p values are now correct throughout the manuscript. In addition, we have clarified the methods section and updated the figure legends to accurately describe the paired nature of our experimental design and the precise statistical tests used each time.

COMMENT 2. With reference to the original point 12, in Fig 4 panel C, the left 2 sets of data now have a stated p -value, with the value of p being 0.2562. This value is quite close to that in panel B, right hand side sets of data, where the p -value is 0.2854, despite the much tighter spread of data in panel B and the much smaller difference in mean values. In view of this, the graph in panel C was blown up in size and the value of each data point was measured. There is of course an element of inaccuracy in such measurements. Nevertheless, an unpaired t -test gives a p -value of ~ 0.07 , which is quite different to the value of 0.2562 in Fig 4C. Paired testing may result in a lower, statistically significant p -value and therefore an effect of U73122 and this would have implications in the interpretation of the data and its discussion. If the data are paired and tested as such, this should be stated in all relevant figure legends.

Response: We thank the reviewer for drawing our attention to this discrepancy. In the previous submission, an unpaired ANOVA was mistakenly applied. The reviewer is correct that if assessed via an unpaired t -test, then the data in Figure 4C (left) gives a p value of 0.07. However, the data in Figure 4B & C was recorded from a single series of experiments performed using a paired (within-subjects) design with $n = 5$ per experiment (each biological replicate serves as its own control). Specifically, basal and then high K^+ -evoked Ca^{2+} responses were first measured. The tissue was then treated with U73122 and basal and high K^+ -evoked Ca^{2+} responses were measured again in the same tissue. The design of this experiment necessitates a repeated measures two-way analysis (with stimulus and treatment as factors), and this has now been performed. This reanalysis has resulted in revised p values and these are now reported in the figures. Importantly, the reanalysis does not affect any of the conclusions of our study.

COMMENT 3. In a similar vein, in Fig 7C left hand graph, the stated p -value is 0.0499, which many readers may consider is too marginal to have any real meaning one way or the other. However, after measuring the individual data points, and given that the data are presented in a paired manner, a paired t -test gives a p -value of 0.006, which is very different to the stated value of 0.0499 and which means that the difference is very significant. Again, the type of statistical testing used should be stated in the figure legend.

Response: Again, we thank the Referee for checking the analysis. The p value previously reported (0.0499, $n = 6$) was for a paired t test. However, on assessing this data for normality of paired differences, we discovered that the assumption of normality was violated. The data has been reassessed using the appropriate Wilcoxon matched-pairs signed rank test, and the correct p value of 0.0312 is now reported.

COMMENT 4. Other p -values in the manuscript were not checked but it is suggested that the authors should check all p -values with great care and include details of the statistical testing in the figure legends.

Response: We thank the Referee for this suggestion. We have reanalysed all the data in the manuscript using the most appropriate paired statistical test for each set of data and report the p values generated throughout. Additionally, we have clarified the methods, results, and figure legends, to clearly indicate the paired design of the experiments and to specify the statistical tests used in each figure.

COMMENT 5. With reference to the original point 2, there is still considerable room for improvement in the English expression.

Response: The manuscript has been edited throughout.

Referee #2:

I am happy with the authors' responses to my main comment but have a couple minor comments on the two other points.

COMMENT 1. The authors stated in their response that in previous study using FaLM did find the organelles [mitochondria] were largely positioned near the plasma membrane (Chalmers et al, 2015, DOI: 10.1038/srep16875). However, that study compared the size, density and clustering of mitochondria in WKY and SHR rats, with no direct evidence that indicates their localisation with the cell membrane. Firth et al, 2009 quantitatively measured co-localisation by labelling the cell membrane, which, in my opinion, doesn't contradict but rather supports the authors' overall conclusion, between SR/mitochondria interactions shown in summary Figure 9.

Response: We have now cited the Firth et al paper in support of the findings of the present study in two separate places in the Discussion section. On Page 20 we have included "In smooth muscle, the SR forms a continuous interconnected network (McCarron & Olson, 2008; Rainbow et al., 2009) that often comes into close proximity with mitochondria. Indeed, in mesenteric artery smooth muscle cells, mitochondria are positioned at distance from the plasma membrane (Firth et al., 2009)."

In the final paragraph of the Discussion section (Page 23) we have included "We also show mitochondria exert little influence on the slow persistent Ca^{2+} rise that occurs as a direct result of voltage-dependent Ca^{2+} entry. This result might be expected given that mitochondria are positioned at distance from the plasma membrane in mesenteric artery smooth muscle cells (Firth et al., 2009)."

COMMENT 2. Animal humane killing: I agree with what the authors said about the choice, but there is no mentioning of a conformational method, I presume exsanguination. This need to me mentioned in the methods.

Response: This has now been clarified in the Methods section (page 7) - animals were euthanised by cervical dislocation, with death confirmed by exsanguination.

Referee #4:

Comments for authors:

The statistical methods section describes the use of paired t-tests and one-way ANOVA for multi-group comparisons, with data reported as mean {plus minus} SD. While the overall approach is broadly appropriate, there are several key concerns that need to be addressed to ensure statistical rigor and the validity of the results:

The methods state that multiple comparisons were conducted following ANOVA but do not specify the correction method used. Without an appropriate post-hoc adjustment (e.g., Bonferroni), there is an increased risk of Type I errors due to multiple hypothesis testing. It is essential to clarify the post-hoc procedure employed and ensure that it appropriately controls for false positives. The authors also need to specify whether their ANOVA was a repeated-measures or independent one-way ANOVA.

Response: We thank the Editor for the careful review of our manuscript. All of our experiments used a paired (before and after) experimental design with responses compared to its own internal control. Our initial submission contained some errors in the statistical analysis, including the use of an unpaired ANOVA for paired data. In this revised submission, we have re-evaluated all data using the appropriate statistical tests (following normality checks) – employing repeated measured ANOVA where applicable. We now specify the exact post-hoc correction methods used (e.g. Dunnett's test to compare each treatment with control). Additionally, we have revised the Methods section to clearly describe the paired experimental design and the specific statistical treatments used.

COMMENT The description indicates that data were extracted from a single preparation with different treatments. This raises the possibility of pseudoreplication if multiple measurements from the same biological sample were treated as independent replicates. It should be explicitly stated whether the replicates represent true biological replicates or technical replicates, as the latter would necessitate statistical adjustments (e.g., mixed-effects models or hierarchical analysis) rather than simple t-tests or ANOVA.

Response: We apologise for any confusion caused by our initial descriptions. To clarify, our experiments were conducted using a paired design. For example, in Figure 2B, tissue was obtained from $n = 5$ separate animals and subjected sequentially to Ca^{2+} removal and then nimodipine treatment. Although multiple Ca^{2+} imaging recordings were taken from a single tissue during these treatments, each recording reflects a different treatment condition, with differences assessed using repeated measures one-way ANOVA (with the Geisser-Greenhouse correction to account for violations of sphericity) followed by Dunnett's multiple comparisons test. We have now explicitly clarified that our sample size for each experiment avoids pseudo replication in the revised manuscript. However, in each experiment, our measurements of Ca^{2+} signalling properties (e.g., amplitude of fast and amplitude of slow response) are independent analyses conducted on the same $n = 5$ biological replicates as is standard practice. This is now also described in the Methods section.

COMMENT The use of a paired t-test assumes normality of differences between paired observations, which is particularly problematic given the small sample size ($n = 5$). With such a low n , normality tests have low power, and violations of normality can substantially impact results. A Wilcoxon signed-rank test would likely be a more robust alternative, providing similar power while avoiding issues related to non-normality. It is recommended that the authors explicitly report whether normality was tested and, if violated, confirm that a non-parametric alternative was used. As above, I would suggest going with non-parametric tests regardless of the outcomes of normality testing, given the very low N . Similarly, if repeated measures ANOVA was used, sphericity should be assessed via Mauchly's test, with Greenhouse-Geisser corrections applied where necessary.

Response: Thank you for these suggestions. We have carefully re-evaluated all data in our manuscript.

All paired differences (or original data for Figure 7G) was assessed for normality and data subsequently assessed using appropriate statistical tests, which are now explicitly stated in each of the figure legends. When data passed the normality test (Shapiro-Wilk test), we used parametric analysis (paired t test, or repeated measures ANOVA, and Welch's t test for Figure 7G). When the data did not pass the normality test, we used non-parametric analysis (Wilcoxon test or Friedman test).

When repeated measures ANOVA was used, the Greenhouse-Geisser correction was applied to account for any violation of sphericity. Full details of the statistical treatments are now provided in the Methods section, and indicate the specific statistical tests used are indicated in each figure legend.

COMMENT While the statistical analysis section states that statistical significance was set at $p < 0.05$, it would be beneficial to supplement this with effect size measures (e.g., Cohen's d for t-tests, η^2 for ANOVA) to provide a better understanding of the magnitude of effects. Useful resource: <https://matthewbjane.quarto.pub/guide-to-effect-sizes-and-confidence-intervals/>. Additionally, GraphPad Prism's default statistical settings should be confirmed, as it may apply corrections automatically without explicit user input.

Response: We appreciate the suggestion and agree that effect size measures can provide valuable additional context. However, incorporating these values directly into the manuscript would require a complete rewrite of the results section and all figure legends. Instead, we have included all original data and test results in a supplementary file, giving interested readers the ability to compute any metrics using from the data. We are unsure if we will be able to include this file as supplementary data with the submission and we are happy, if that is the case, to upload the file to a data repository (e.g. figshare) if needed. Additionally, we confirmed the GraphPad Prism settings used in all analyses.

Dear Dr McCarron,

Re: JP-RP-2025-288022R2 "Mitochondria regulate IP3-mediated Ca²⁺ release triggered by voltage-dependent Ca²⁺ entry in resistance arteries" by Xun Zhang, Charlotte Buckley, Matthew D Lee, Susan Chalmers, Calum Wilson, and John G McCarron

Thank you for submitting your manuscript to The Journal of Physiology. It has been assessed by a Reviewing Editor and by 2 expert referees and we are pleased to tell you that it is acceptable for publication following satisfactory revision.

REVISION CHECKLIST:

We look forward to receiving your revised submission.

Yours sincerely,

Bjorn Knollmann
Senior Editor
The Journal of Physiology

REQUIRED ITEMS

- Papers must comply with the Statistics Policy: https://jp.msubmit.net/cgi-bin/main.plex?form_type=display_requirements#statistics.

In summary:

- If $n \leq 30$, all data points must be plotted in the figure in a way that reveals their range and distribution. A bar graph with data points overlaid, a box and whisker plot or a violin plot (preferably with data points included) are acceptable formats.
- If $n > 30$, then the entire raw dataset must be made available either as supporting information, or hosted on a not-for-profit repository, e.g. FigShare, with access details provided in the manuscript.
- 'n' clearly defined (e.g. x cells from y slices in z animals) in the Methods. Authors should be mindful of pseudoreplication.
- All relevant 'n' values must be clearly stated in the main text, figures and tables.
- The most appropriate summary statistic (e.g. mean or median and standard deviation) must be used. Standard Error of the Mean (SEM) alone is not permitted.
- Exact p values must be stated. Authors must not use 'greater than' or 'less than'. Exact p values must be stated to three significant figures even when 'no statistical significance' is claimed.

EDITOR COMMENTS

Reviewing Editor:

Just a few minor editorial corrections. Great study

Senior Editor:

Comments for Authors to ensure the paper complies with the Statistics Policy:

Please state exact p-values and provide author contributions

The revision is much improved and only a few editorial corrections are missing before the article can be accepted. Please make sure the final MS follows our statistical guidelines

REFEREE COMMENTS

Referee #1:

1. The authors have now further revised the manuscript, and in so doing they have incorporated re-analysis of the statistics. The authors have therefore satisfactorily addressed the issues that were raised previously, thereby improving the manuscript. There are some minor editing issues that the authors could address, as indicated below.

2. In Fig 8C the Y-axis has units of area. As indicated in the original review, the units of length (metre) are abbreviated with lower case "m", not the upper case "M" that is in the figure. [For information, "M" is the unit for Molarity]

3. Line 431 ends with "In contrast," Either the remainder of the sentence is missing or the sentence has been incompletely deleted.

4. In line 122, "organelles" requires a possessive apostrophe, thus "organelles' ".

5. In several places the wording "Depolarizing the mitochondrial membrane potential ..." or similar, is used. Strictly speaking, it is the membrane that is depolarized, not the potential (which is decreased). Thus, it would be better to omit "potential" and just state "Depolarizing the mitochondrial membrane".

6. In the References section, whereas most journal names are given in full, the names of journals are abbreviated or incomplete for 11 references.

Referee #2:

Revision was performed satisfactorily, no further comments

END OF COMMENTS

Dr Bjorn Knollmann
Senior Editor
The Journal of Physiology

Dear Dr Knollmann

Thank you, the Editors and the Referees for your time and consideration of our manuscript.

In the rebuttals below, detailed responses to each point raised by the Editors and Reviewers are provided and we explain the changes made in the revised manuscript to address the points raised.

Once again, we would like to express our gratitude for the opportunity to submit our work to Journal of Physiology and for the helpful feedback provided by the Editor and Reviewers.

Yours Sincerely,

John G McCarron
W.C. Bowman Chair of Pharmacology

COMMENT - Papers must comply with the Statistics Policy: https://jp.msubmit.net/cgi-bin/main.plex?form_type=display_requirements#statistics.

- If $n \leq 30$, all data points must be plotted in the figure in a way that reveals their range and distribution. A bar graph with data points overlaid, a box and whisker plot or a violin plot (preferably with data points included) are acceptable formats.

Response: All data point are shown in each figure.

COMMENT - If $n > 30$, then the entire raw dataset must be made available either as supporting information, or hosted on a not-for-profit repository, e.g. FigShare, with access details provided in the manuscript.

Response: Not applicable.

COMMENT - 'n' clearly defined (e.g. x cells from y slices in z animals) in the Methods. Authors should be mindful of pseudoreplication.

Response: The reported n value is the number of biological replicates (i.e., the number of animals) as outlined in the Methods section

COMMENT - All relevant 'n' values must be clearly stated in the main text, figures and tables.

Response: All relevant n values, which represent the number of independent biological replicates (animals), are clearly states in the main text, as well as in the figure legends. Additionally, individual data points are shown in figures.

COMMENT - The most appropriate summary statistic (e.g. mean or median and standard deviation) must be used. Standard Error of the Mean (SEM) alone is not permitted.

Response: Mean and standard deviation of n independent biological replicates is used throughout.

COMMENT - Exact p values must be stated. Authors must not use 'greater than' or 'less than'. Exact p values must be stated to three significant figures even when 'no statistical significance' is claimed.

Response: All p values are reported in the respective figures to three significant figures. All data and test results are included in the supplementary data.

EDITOR COMMENTS

Reviewing Editor:

Just a few minor editorial corrections. Great study

Response: We thank the Editor for the comments

Senior Editor:

Comments for Authors to ensure the paper complies with the Statistics Policy:

Please state exact p-values and provide author contributions

Response: Done All p values are reported in the respective figures to three significant figures. All data and test results are included in the supplementary data. Author contributions are included in the "Additional Information" section.

The revision is much improved and only a few editorial corrections are missing before the article can be accepted. Please make sure the final MS follows our statistical guidelines

Response: We thank the Editor for the comments. Statistical guidelines are followed throughout (see response above) and the editorial corrections requested have been made.

REFeree COMMENTS

Referee #1:

COMMENT 1. The authors have now further revised the manuscript, and in so doing they have incorporated re-analysis of the statistics. The authors have therefore satisfactorily addressed the issues that were raised previously, thereby improving the manuscript. There are some minor editing issues that the authors could address, as indicated below.

Response: We thank the Referee for the careful attention and helpful comments

COMMENT 2. In Fig 8C the Y-axis has units of area. As indicated in the original review, the units of length (metre) are abbreviated with lower case "m", not the upper case "M" that is in the figure. [For information, "M" is the unit for Molarity]

Response: Corrected. Unit is changed to "µm" in figure 8C

COMMENT 3. Line 431 ends with "In contrast," Either the remainder of the sentence is missing or the sentence has been incompletely deleted.

Response: The sentence was incompletely deleted and 'In contrast' has been removed.

COMMENT 4. In line 122, "organelles" requires a possessive apostrophe, thus "organelles' ".

Response: Corrected - "organelles" is changed to "organelles' " in the text

COMMENT 5. In several places the wording "Depolarizing the mitochondrial membrane potential ..." or similar, is used. Strictly speaking, it is the membrane that is depolarized, not the potential (which is decreased). Thus, it would be better to omit "potential" and just state "Depolarizing the mitochondrial membrane".

Response: "mitochondrial membrane potential" has been changed to "mitochondrial membrane" where is appropriate in the text.

COMMENT 6. In the References section, whereas most journal names are given in full, the names of journals are abbreviated or incomplete for 11 references.

Response: Corrected - All journals are now given in full name in the reference section.

Referee #2:

COMMENT Revision was performed satisfactorily, no further comments

Response: We thank the Referee for the careful attention and helpful comments

END OF COMMENTS

Dear Dr McCarron,

Re: JP-RP-2025-288022R3 "Mitochondria regulate IP3-mediated Ca²⁺ release triggered by voltage-dependent Ca²⁺ entry in resistance arteries" by Xun Zhang, Charlotte Buckley, Matthew D Lee, Susan Chalmers, Calum Wilson, and John G McCarron

We are pleased to tell you that your paper has been accepted for publication in The Journal of Physiology.

Yours sincerely,

Bjorn Knollmann
Senior Editor
The Journal of Physiology

If you would like to receive our 'Research Roundup', a monthly newsletter highlighting the cutting-edge research published in The Physiological Society's family of journals (The Journal of Physiology, Experimental Physiology, Physiological Reports, The Journal of Nutritional Physiology and The Journal of Precision Medicine: Health and Disease), please click this link, fill in your name and email address and select 'Research Roundup':
<https://www.physoc.org/journals-and-media/membernews>

- You can help your research get the attention it deserves! Check out Wiley's free Promotion Guide for best-practice recommendations for promoting your work at: www.wileyauthors.com/eoo/guide. You can learn more about Wiley Editing Services which offers professional video, design, and writing services to create shareable video abstracts, infographics, conference posters, lay summaries, and research news stories for your research at: www.wileyauthors.com/eoo/promotion.

EDITOR COMMENTS

Reviewing Editor:

I have no further comments

Senior Editor:

The work is now acceptable for publication. Thank you for contributing this excellent paper!